# A FRAMEWORK FOR LEARNED COUNTSKETCH

## ABSTRACT

Sketching is a compression technique that can be applied to many problems to solve them quickly and approximately. The matrices used to project data to smaller dimensions are called "sketches". In this work, we consider the problem of optimizing sketches to obtain low approximation error over a data distribution.

We introduce a general framework for "learning" and applying *CountSketch*, a type of sparse sketch. The sketch optimization procedure has two stages: one for optimizing the placements of the sketch's non-zero entries and another for optimizing their values. Next, we provide a way to apply learned sketches that has worst-case guarantees for approximation error.

We instantiate this framework with three sketching applications: least-squares regression, low-rank approximation (LRA), and $k$-means clustering. Our experiments demonstrate that our approach substantially decreases approximation error compared to classical and naïvely learned sketches.

Finally, we investigate the theoretical aspects of our approach. For regression and LRA, we show that our method obtains state-of-the art accuracy for fixed time complexity. For LRA, we prove that it is strictly better to include the first optimization stage for two standard input distributions. For $k$-means, we derive a more straightforward means of retaining approximation guarantees.

## 1 INTRODUCTION

In recent years, we have seen the influence of machine learning extend far beyond the field of artificial intelligence. The underlying paradigm, which assumes that a given algorithm has an input *distribution* for which algorithm parameters can be optimized, has even been applied to classical algorithms. Examples of classical problems that have benefitted from ML include cache eviction strategies, online algorithms for job scheduling, frequency estimation of data stream elements, and indexing strategies for data structures (Lykouris & Vassilvitskii, 2018; Purohit et al., 2018; Hsu et al., 2019; Kraska et al., 2018). This input distribution assumption is often realistic. For example, many real-world applications use data streaming to track things like product purchasing statistics in real time. Consecutively streamed datapoints are usually tightly correlated and closely fit certain distributions.

We are interested in how this distributional paradigm can be applied to sketching, a data compression technique. With the dramatic increase in the dimensions of data collected in the past decade, compression methods are more important than ever. Thus, it is of practical interest to improve the accuracy and efficiency of sketching algorithms.

We study a sketching scheme in which the input matrix is compressed by multiplying it with a "sketch" matrix with a small dimension. This smaller, sketched input is then used to compute an approximate solution. Typically, the sketch matrix and the approximation algorithm are designed to satisfy *worst-case* bounds on approximation error for *arbitrary* inputs. With the ML perspective in mind, we examine if it is possible to construct sketches which also have low error in *expectation* over an input distribution. Essentially, we aim for the best of both worlds: good performance in practice *with* theoretical worst-case guarantees.

Further, we are interested in methods that work for multiple sketching applications. Typically, sketching is very application-specific. The sketch construction and approximation algorithm are tailored to individual applications, like robust regression or clustering (Sarlos, 2006; Clarkson & Woodruff, 2009; 2014; 2017; Cohen et al., 2015; Makarychev et al., 2019). Instead, we consider

three applications at once (regression, LRA, $k$-means) and propose generalizable methods, as well as extending previous application-specific work.

**Our results.** At a high level, our work's aim is to make sketch learning more *effective*, *general*, and ultimately, *practical*. We propose a framework for constructing and using learned CountSketch. We chose CountSketch because it is a sparse, input-independent sketch (Charikar et al., 2002). Specifically, it has one non-zero entry ($\pm 1$) per column and does not need to be constructed anew for each input matrix it is applied to. These qualities enable CountSketch to be applied quickly, since sparse matrix multiplication is fast and we can reuse the same CountSketch for different inputs. Our "learned" CountSketch will retain this characteristic sparsity pattern and input-independence[1], but its non-zero entries will range in $\mathbb{R}$.

We list our main contributions and follow this with a discussion.

- **Two-stage sketch optimization**: to first place the non-zero entries and then learn their values.
- **Theoretical worst-case guarantees, two ways**: we derived a *time-optimal* method which applies to MRR, LRA, $k$-means, and more. We also proved a simpler method works for $k$-means.
- **SOTA experimental results**: we showed the versatility of our method on 5 data sets with 3 types. Our method dominated on the majority of experiments.
- **Theoretical analysis on the necessity of two stages**: we proved that including the first stage is strictly better for LRA and two common input distributions.
- **Empirical demonstration of the necessity of two stages**: showed that including the first stage gives a $12, 20\%$ boost for MRR, LRA.

Our sketch learning algorithm first places the sparse non-zero entries using a greedy strategy, and then learns their values using gradient descent. The resulting learned CountSketch is very different from the classical CountSketch: the non-zero entries no longer have random positions and $\pm 1$ values. As a result, the usual worst-case guarantees do not hold.

We sought a way to obtain worst-case guarantees that was fast and reasonably general. Our solution is a *fast comparison step* which performs an approximate evaluation of learned and classical sketches and takes the better of the two. Importantly, we can run this step before the approximation algorithm without increasing its overall time complexity. As such, this solution is *time-optimal* and applies to MRR, LRA, $k$-means, and more.

An alternate method was proposed by a previous work, but it was only proved for LRA (Indyk et al., 2019). This "sketch concatenation" method just involves sketching with the concatenation of a learned and a classical sketch. Since it is somewhat simpler, we wanted to extend its applicability. In a novel theoretical result, we proved this works for $k$-means as well.

We also ran a diverse set of experiments to demonstrate the versatility and practicality of our approach. We chose five data sets spanning three categories (image, text, and graph) to test our method on three applications (MRR, LRA, $k$-means). Importantly, these experiments have real-world counterparts. For example, LRA and $k$-means can be used to compress images, applying SVD (LRA) to text data is the basis of a natural language processing technique, and LRA can be used to compute approximate max cuts on graph adjacency matrices. Ultimately, our method dominated on the vast majority of tests, giving a $31, 70\%$ improvement over classical CountSketch for MRR, LRA.

Finally, we conducted ablation study of the components of our algorithm. In another novel theoretical result, we proved that including the time-consuming first optimization stage is strictly better than not to for LRA and two input distributions (spiked covariance and Zipfian). Empirically, this is case for all 3 applications.

**Related work.** In the last few years, there has been much work on leveraging ML to improve classical algorithms; we only mention a few examples here. One related body of work is data-dependent

---

[1]While learned CountSketch is data-dependent (it is optimized using sample input matrices), it is still considered input-independent because it is applied to unseen input matrices (test samples).

dimensionality reduction, such as an approach for pair-wise/multi-wise similarity preservation for indexing big data (Wang et al., 2017) and a method for learning linear projections for general applications (Hegde et al., 2015). We note that multiplying an input matrix on the left with a sparse sketch is equivalent to hashing its rows to a small number of bins. Thus, we also find connections with the body of work on learned hashes, most of which addresses the nearest neighbor search problem (see Wang et al. for a survey). However, in order to obtain approximation guarantees, our "hash function" (sparse sketch) must satisfy properties which these learned hashes usually do not, such as affine $\epsilon$-embedding (Def. A.1).

In particular, we build off of the work of Indyk et al. (2019), which introduced gradient descent optimization for the LRA application. It also gave an LRA-specific method for worst-case guarantees. We have surpassed the sketching performance and breadth of this work. Namely, we introduce a *sparsity pattern* optimization step which is clearly crucial for *sparse* sketches. We also provide a more general method for worst-case guarantees and extend their method to $k$-means.

## 2 PRELIMINARIES

Our learned sketches have the sparsity pattern of the classical CountSketch. The construction of this sketch is described below. We also define affine $\epsilon$-embeddings, which is a class of sketches that includes CountSketch. This $\epsilon$-embedding property is desirable because it allows us to prove that certain sketching algorithms give $(1 + \epsilon)$-approximations.

**Definition 2.1 (Classical CountSketch).** *The* CountSketch *(abbreviated as CS) matrix S has one non-zero entry in each column with a random location and random value in* $\{\pm 1\}$*.*

**Definition 2.2 (Affine Embedding).** *Given a pair of matrices $A$ and $B$, a matrix $S$ is an* affine $\epsilon$-embedding *if for all $X$ of the appropriate shape,* $\|S(AX - B)\|_F^2 = (1 \pm \epsilon) \|AX - B\|_F^2$*.*

**Notation.** We denote the singular value decomposition (SVD) of $A$ by $A = U\Sigma V^\top$ with orthogonal $U, V$ and diagonal $\Sigma$. Relatedly, the Moore-Penrose pseudo-inverse of $A$ is $A^\dagger = V\Sigma^{-1}U^\top$, where $\Sigma^{-1}$ is constructed by inverting the non-zero diagonal entries.

## 3 FRAMEWORK

We describe a framework for learned CountSketch that can be adopted by many different applications, including least-squares regression (MRR), low-rank approximation (LRA), and $k$-means clustering. We will return to these applications in the next section. In this section, we first describe how to optimize a CountSketch over a set of training samples. Then, we explain how to use the learned CountSketch to achieve good *expected* performance with *worst-case* guarantees. By running a "fast comparison" step before the approximation algorithm, we can do this in optimal time.

### SKETCH OPTIMIZATION

Most applications are optimization problems. That is, they are defined by an objective function, $\mathcal{L}(\cdot)$. For example, least-squares regression solves

$$\min_X \|AX - B\|_F^2$$

given $A \in \mathbb{R}^{n \times d}, B \in \mathbb{R}^{n \times d'}$. Of course, the optimal solution is a function of the inputs:

$$X^* = \arg\min_X \|AX - B\|_F^2 = A^\dagger B$$

However, in the sketching paradigm, we compute an *approximately optimal* solution as a function of the *sketched* (compressed) inputs. Taking CountSketch $S \in \mathbb{R}^{m \times n}$ for $m \ll n$, we have:

$$\hat{X}^* = (SA)^\dagger (SB)$$

Our goal is to minimize the expected approximation error with respect to $S$, which is constrained to the set of CountSketch-sparse matrices ($\mathcal{CS}$). However, this is simply equivalent to minimizing the

objective value of the approximate solution.

$$S^* = \arg\min_{S \in \mathcal{CS}} \mathbb{E}_{(A,B) \sim \mathcal{D}} \left[ \mathcal{L}_{A,B}((SA)^\dagger(SB)) - \mathcal{L}_{A,B}(X^*) \right]$$

$$= \arg\min_{S \in \mathcal{CS}} \mathbb{E}_{(A,B) \sim \mathcal{D}} \left[ \mathcal{L}_{A,B}((SA)^\dagger(SB)) \right]$$

For ease of notation, we will define $G(\cdot)$ as a function which maps a sketch and inputs to an approximate solution. $G(\cdot)$ is defined by the application-specific approximation algorithm. For MRR, $G(S, (A, B)) = (SA)^\dagger(SB)$. More generally, the sketch optimization objective is:

$$S^* = \arg\min_{S \in \mathcal{CS}} \mathbb{E}_{A \sim \mathcal{D}} \left[ \mathcal{L}_A(G(S, A)) \right] \tag{3.1}$$

If the application is regression, we let $A$ be $(A, B)$.

We will solve this constrained optimization in two stages. For both stages, we approximate the expectation in empirical risk minimization (ERM) fashion. That is, we approximate the expectation over the true distribution by averaging over a sampled batch of the training set. Now, in the first optimization stage, we compute positions for the CountSketch-sparse nonzero entries. Then, in the second stage, we fix the positions and optimize the nonzero values.

**Stage 1: Placing the nonzero entries.** We want to maintain the sparsity pattern of CS (one nonzero entry per column), but we are free to place that nonzero entry wherever we like for each column. A naïve method would be to evaluate the objective for the exponential number of full placements. This is clearly intractable, so we consider a greedy alternative. In essence, we construct the sketch one nonzero entry at a time, and we choose the location of the next entry by minimizing (3.1) over the discrete set of possibilities.

More precisely, we build the sketch $S \in \mathbb{R}^{m \times n}$ iteratively, placing one nonzero entry at a time. For each nonzero entry, we consider $m$ locations and 2 values for each location ($\pm 1$). We evaluate the sketch optimization objective (3.1) for all $2m$ incremental updates to $S$ and choose the minimizing update. In the pseudo-code below, we iterate through the $n$ columns of $S$, each of which contains a non-zero entry. Note that $S_{w,j} = S + w(\vec{e_j}\vec{e_i}^\top)$ adds a single entry $w$ to the $i$-th column, $j$-th row of the current, partially-constructed $S$.

---

**Algorithm 1** GREEDY STAGE

**Require:** $\mathcal{A}_{\text{train}} = \{A_1, ..., A_N\}$ with $A_i \in \mathbb{R}^{n \times d}$; sketch dimension $m$
1: **initialize** $S = \mathbb{0}^{m \times n}$
2: **for** $i = 1$ to $n$ **do**
3:   $w^*, j^* = \arg\min_{w \in \{\pm 1\}, j \in [m]} \sum_{A_i \in \mathcal{A}_{\text{train}}} \mathcal{L}_{A_i}(G(S_{w,j}, A_i))$ where $S_{w,j} = S + w(\vec{e_j}\vec{e_i}^\top)$
4:   $S[j^*, i] = w^*$
5: **end for**

---

For some applications it can be inefficient to evaluate (3.1), since it requires computing the approximate solution. For MRR and LRA, the approximate solution has a closed form, but for $k$-means, it must be computed iteratively. This is prohibitively expensive, since we perform many evaluations. In this case, we recommend finding a surrogate $\mathcal{L}(\cdot)$ with a closed-form solution, as we illustrate in later sections.

**Stage 2: Optimizing the nonzero values.** We now fix the positions of the nonzero entries and optimize their values using gradient descent. To fix the positions, we represent $S$ as just a vector of its nonzero entries, $\vec{v} \in \mathbb{R}^n$. We will denote $H(\vec{v}) : \mathbb{R}^n \to \mathbb{R}^{m \times n}$ as the function which maps this concise representation of $S$ to the full matrix. $H(\cdot)$ depends on the positions computed in the last stage, which are fixed.

Now, we simply differentiate $\mathbb{E}_{A \sim \mathcal{D}} \left[ \mathcal{L}_A(G(H(\vec{v}), A)) \right]$ (3.1) with respect to $\vec{v}$.

---

**Algorithm 2** GRADIENT STAGE

---

**Require:** $\mathcal{A}_{\text{train}} = \{A_1, ..., A_N\}$ with $A_i \in \mathbb{R}^{n \times d}$; $H(\cdot)$ from Alg. 1; learning rate $\alpha$
1: **for** $i = 1$ to $n_{\text{iter}}$ **do**
2:     $S = \mathbb{0}^{m \times n}$
3:     **sample** $\mathcal{A}_{\text{batch}}$ from $\mathcal{A}_{\text{train}}$
4:     $\vec{v} \leftarrow \vec{v} - \alpha \left( \displaystyle\sum_{A \in \mathcal{A}_{\text{batch}}} \frac{\partial \mathcal{L}_A(G(H(\vec{v})), A)}{\partial \vec{v}} \right)$
5: **end for**

---

LEARNED SKETCH WITH WORST-CASE GUARANTEES

We first run a *fast comparison* between our learned sketch and a classical one and then run the approximation algorithm with the "winner". This allows us to compute an approximate solution (to the applications we consider here; i.e., MRR and LRA) that does not perform worse than classical CountSketch. In other words, our solution has the same worst-case guarantees as classical CountSketch. The benefit of this is that these guarantees hold for *arbitrary* inputs, so our method is protected from out-of-distribution inputs as well as in-distribution inputs which were not well-represented in the training data.

More precisely, for a given input $A$, we *quickly* compute a *rough estimate* of the approximation errors for learned and classical CountSketches. This rough estimate can be obtained by sketching. We take the sketch with the better approximation error and use it to run the usual approximation algorithm.

The choice to compute a *rough estimate* of the approximation error rather than the exact value is crucial here. It allows us to append this *fast comparison* step without increasing the time complexity of the approximation algorithm. Thus, the whole procedure is **still time-optimal**.

Though this method is simple, an even simpler method exists for some applications. Indyk et al. proved that "sketch concatenation" (sketching with a classical sketch appended to the learned one) retains worst-case guarantees for LRA. We prove that this also works for $k$-means clustering (Theorem 4.6).

---

**Algorithm 3** LEARNED-SKETCH-ALGORITHM

---

**Require:** learned sketch $S_L$; classical sketch $S_C$; trade-off parameter $\beta$; input data $A$
1: Define $M_L, M_C$ such that $\mathcal{L}_A(G(S_L, A)) = \|M_L\|_F^2$, $\mathcal{L}_A(G(S_C, A)) = \|M_C\|_F^2$
2: $S \leftarrow$ CountSketch $\in \mathbb{R}^{\frac{1}{\beta^2} \times n}$, $R \leftarrow$ CountSketch $\in \mathbb{R}^{\frac{1}{\beta^2} \times d}$
3: $\Delta_L \leftarrow \|SM_LR^\top\|_F^2$, $\Delta_C \leftarrow \|SM_CR^\top\|_F^2$
4: **if** $\Delta_L \leq \Delta_C$ **then**
5:     **return** $G(S_L, A)$
6: **end if**
7: **return** $G(S_C, A)$

---

This algorithm can be used for applications that minimize a Frobenius norm. In the MRR example,

$$\mathcal{L}_A(G(S_L, A)) = \left\| A \left[ (S_L A)^\dagger (SB) \right] - B \right\|_F^2$$

so $M_L = A \left[ (S_L A)^\dagger (SB) \right] - B$. Note that $\beta$ is a parameter which trades off the precision of the approximation error estimate and the runtime of Algorithm 3.

## 4   INSTANTIATIONS

For each of 3 problems (least-squares regression, low-rank approximation, $k$-means clustering), we define the problem's objective, $\mathcal{L}_A(\cdot)$, and the approximation algorithm, $G(\cdot)$, which maps a sketch and inputs $(S, A)$ to an approximate minimizer of $\mathcal{L}_A(\cdot)$.

### 4.1 LEAST-SQUARES REGRESSION

We consider a generalized version of $\ell_2$ regression called "multiple-response regression" (MRR). Given a matrix of observations ($A \in \mathbb{R}^{n \times d}$, with $n \gg d$) and a matrix of corresponding values ($B \in \mathbb{R}^{n \times d'}$, with $n \gg d'$), the goal of MRR is to solve

$$\min_X \mathcal{L}_{(A,B)}(X) = \min_X \|AX - B\|_F^2$$

**Approximation algorithm.** The algorithm is simply to sketch $A, B$ with a sparse sketch $S$ (e.g., CS) and compute the closed-form solution on $SA, SB$, which are small (Algorithm 4).

---

**Algorithm 4** SKETCH-REGRESSION (Sarlos, 2006; Clarkson & Woodruff, 2017)

**Require:** $A \in \mathbb{R}^{n \times d}, B \in \mathbb{R}^{n \times d'}, S \in \mathbb{R}^{m \times n}$
  1: **return:** $(SA)^\dagger (SB)$

---

**Sketch optimization.** For both the greedy and gradient stages, we use the objective $\mathcal{L}_{(A,B)}(G(S,(A,B))) = \|A(SA)^+(SB) - B\|_F^2$.

**Learned sketch algorithm.** For fixed accuracy, our learned sketch algorithm achieves state-of-the-art time complexity, besting the classical algorithm. We prove an equivalent statement: the learned sketch algorithm yields better approximation error for fixed time complexity. (See Lemma A.4 for the classical algorithm's worst-case guarantee). In the following theorem, we assume Alg. 3 uses a learned sketch which is an affine $\beta$-embedding and also a classical sketch of the same size which is an affine $\epsilon$-embedding. If $\beta < \epsilon$, the above statement is true.

**Theorem 4.1 (Learned sketching with guarantees for MRR).** *Given a learned sparse sketching matrix $S_L \in \mathbb{R}^{\frac{d^2}{\epsilon^2} \times n}$ which attains a $(1 + \gamma)$-approximation for MRR on $A \in \mathbb{R}^{n \times d}, B \in \mathbb{R}^{n \times d'}$, Alg. 3 gives a $(1 + O(\beta + \min(\gamma, \epsilon)))$-approximation for MRR on $A$ in $O(\mathrm{nnz}(A) + \mathrm{nnz}(B) + d^5 d' \epsilon^{-4} + d\beta^{-4})$ time where $\beta$ is a trade-off parameter.*

**Remark 4.2.** By setting the trade-off parameter $\beta^{-4} = O(\epsilon^{-4} d^4 d')$, Alg. 3 has the same asymptotic runtime as the best $(1 + \epsilon)$-approximation algorithm of MRR with classical sparse embedding matrices. Moreover, Alg. 3 for MRR achieves a strictly better approximation bound $(1 + O(\beta + \gamma)) = (1 + o(\epsilon))$ when $\gamma = o(\epsilon)$. On the other hand, in the worst case scenario when the learned sketch performs poorly (i.e., $\gamma = \Omega(\epsilon)$) the approximation guarantee of Alg. 3 remains $(1 + O(\epsilon))$.

### 4.2 LOW-RANK APPROXIMATION

Given an input matrix $A \in \mathbb{R}^{n \times d}$ and a desired rank $k$, the goal of LRA is to solve

$$\min_{X, \text{rank } k} \mathcal{L}_A(X) = \min_{X, \text{rank } k} \|X - A\|_F^2$$

**Approximation algorithm.** We consider the time-optimal (up to low order terms) approximation algorithm by Sarlos; Clarkson & Woodruff; Avron et al. (Algorithm 5) with worst-case guarantees described in Lemma A.8.

---

**Algorithm 5** SKETCH-LOWRANK (Sarlos, 2006; Clarkson & Woodruff, 2017; Avron et al., 2016).

**Require:** $A \in \mathbb{R}^{n \times d}, S \in \mathbb{R}^{m_S \times n}, R \in \mathbb{R}^{m_R \times d}, S_2 \in \mathbb{R}^{m_{S_2} \times n}, R_2 \in \mathbb{R}^{m_{R_2} \times d}$
  1: $U_C \begin{bmatrix} T_C & T_C' \end{bmatrix} \leftarrow S_2 A R^\top, \begin{bmatrix} T_D^\top \\ T_D'^\top \end{bmatrix} U_D^\top \leftarrow SAR_2^\top$ with $U_C, U_D$ orthogonal
  2: $G \leftarrow S_2 A R_2^\top$
  3: $Z_L' Z_R' \leftarrow [U_C^\top G U_D]_k$
  4: $Z_L = \begin{bmatrix} Z_L' T_D^{-\top} & 0 \end{bmatrix}, Z_R = \begin{bmatrix} T_C^{-1} Z_R' \\ 0 \end{bmatrix}$
  5: $Z = Z_L Z_R$
  6: **return:** $AR^\top ZSA$ in form $P_{n \times k}, Q_{k \times d}$

---

**Sketch optimization.** For the greedy stage, we optimize sketches $S$ and $R$ individually. However, we do not use $\mathcal{L}_A(G((S, R, S_2, R_2), A)) = \|X - A\|_F^2$ with $X$ from Alg. 5 as the objective. This is because the optimization for one sketch would then depend on the other sketches. Instead, we use a proxy objective. We observe that the proof that Alg. 5 is an $\epsilon$-approximation uses the fact that the row space of $SA$ and the column space of $AR$ both contain a good rank-$k$ approximation to $A$. Thus, the proxy objectives for $S$ and $R$ are for $k$-rank approximation in the row/column space of $SA/AR$, respectively. For example, the proxy objective for $S$ is $\mathcal{L}_A(G(S, A)) = \left\|[AV]_k V^\top - A\right\|_F^2$ where $V$ is from the SVD $SA = U\Sigma V^\top$ and $[\cdot]_k$ takes the optimal $k$-rank approximation using truncated SVD. The proxy objective for $R$ is defined analogously.

For the gradient stage, we optimize all four sketches $(S, R, S_2, R_2)$ simultaneously using $\mathcal{L}_A(G((S, R, S_2, R_2), A)) = \|X - A\|_F^2$ with $X$ from Algorithm 5, since it can easily be implemented with differentiable functions.

**Learned sketch algorithm.** Just like for regression (Section 4.1), we can prove that our learned sketch algorithm achieves a better accuracy than the classical one given the same runtime.

**Theorem 4.3 (Low-Rank Approximation).** *Given learned sparse sketching matrix* $S_L \in \mathbb{R}^{\mathrm{poly}(\frac{k}{\epsilon}) \times n}$, $R_L \in \mathbb{R}^{\mathrm{poly}(\frac{k}{\epsilon}) \times d}$ *which attain a* $(1 + \gamma)$-*approximation for LRA on* $A \in \mathbb{R}^{n \times d}$, *Alg. 3 gives a* $(1 + \beta + \min(\gamma, \epsilon))$-*approximation for LRA on* $A$ *in* $O(\mathrm{nnz}(A) + (n + d)\,\mathrm{poly}(\frac{k}{\epsilon}) + \frac{k^4}{\beta^4} \cdot \mathrm{poly}(\frac{k}{\epsilon}))$ *time where* $\beta$ *is a trade-off parameter.*

**Remark 4.4.** For $\epsilon \gg k(n+d)^{-4}$, by setting the trade-off parameter $\beta^{-4} = O(k(n+d)^{-4})$, Alg. 3 has the same asymptotic runtime as the best $(1 + \epsilon)$-approximation algorithm of LRA with classical sparse embedding matrices. Moreover, Alg. 3 for LRA achieves a strictly better approximation bound $(1 + O(\beta + \gamma)) = (1 + o(\epsilon))$ when $\gamma = o(\epsilon)$. On the other hand, in the worst case scenario when the learned sketches perform poorly (i.e., $\gamma = \Omega(\epsilon)$) the approximation guarantee of Alg. 3 remains $(1 + O(\epsilon))$.

**Greedy stage offers strict improvement.** Finally, we prove for LRA that including the greedy stage is strictly better than not including it. We can show this for two different natural distributions (spiked covariance and Zipfian). The intuition is that the greedy algorithm separates heavy-norm rows (which are important "directions" in the row space) into different bins.

**Theorem 4.5.** *Consider a matrix* $A$ *from either the spiked covariance or Zipfian distribution. Let* $S_L$ *denote a sparse sketch that Algorithm 1 has computed by iterating through indices in order of non-increasing row norms of* $A$. *Let* $S_C$ *denote a CountSketch matrix. Then, there is a fixed* $\eta > 0$ *such that* $\min_{\text{rank-}k\ X \in \mathrm{rowsp}(S_L A)} \|X - A\|_F^2 \le (1 - \eta) \min_{\text{rank-}k\ X \in \mathrm{rowsp}(S_C A)} \|X - A\|_F^2$.

### 4.3 $k$-MEANS CLUSTERING

Let $A \in \mathbb{R}^{n \times d}$ represent a set of $n$ points, $A_1, \dots, A_n \in \mathbb{R}^d$. In $k$-means clustering, the goal is to find a partition of $A_1, ..., A_n$ into $k$ clusters $\mathcal{C} = \{C_1, \dots, C_k\}$ to

$$\min_{\mathcal{C}} \mathcal{L}_A(\mathcal{C}) = \min_{\mathcal{C}} \sum_{i=1}^{k} \min_{\mu_i \in \mathbb{R}^d} \sum_{j \in C_i} \|A_j - \mu_i\|_F^2$$

where $\mu_i$ denotes the center of cluster $C_i$.

**Approximation algorithm.** First, compress $A$ into $AV$, where $V$ is from SVD $SA = U\Sigma V^\top$ and $S \in \mathbb{R}^{O(k^2/\epsilon^2) \times n}$ is a CountSketch. Then, we use an approximation algorithm for $k$-means, such as Lloyd's algorithm with a $k$-means++ initialization (Lloyd, 1982; Arthur & Vassilvitskii, 2007). Solving with $AV$ gives a $(1+\epsilon)$-approximation (Cohen et al., 2015) and Lloyd's gives an $O(\log(k))$ approximation, so we have a $(1 + \epsilon)O(\log(k))$ approximation overall.

**Sketch optimization.** $k$-means is an interesting case study because it presents a challenge for both stages of optimization. For the greedy stage, we observe that $k$-means does not have a closed form solution, which means that evaluating the objective requires a (relatively) time-consuming iterative process. In the gradient stage, it is possible to differentiate through this iterative computation,

but propagating the gradient through such a nested expression is time-consuming. Thus, a proxy objective which is simple and quick to evaluate would be useful for both stages.

Cohen et al. showed that we get a good $k$-means solution if $A$ is projected to an *approximate top singular vector space*. This suggests that a suitable proxy objective would be low-rank approximation: $\mathcal{L}_A(G(S, A)) = \left\|[AV]_k V^\top - A\right\|_F^2$, like in Section 4.2. We use this objective for both the greedy and gradient stages.

**Learned sketch algorithm.** For $k$-means, we can use a different method to obtain worst-case guarantees. We prove that by concatenating a classical sketch to our learned sketch, our sketched solution will be a $(1 + O(\epsilon))$-approximation.

**Theorem 4.6 (Sketch monotonicity property for $k$-means).** *For a given classical CountSketch $S_C \in \mathbb{R}^{O(\mathrm{poly}(k/\epsilon)) \times n}$, sketching with any extension of $S_C$ (i.e., by a learned sparse sketch $S_L$) yields a better approximate solution for $k$-means than sketching with $S_C$ itself.*

## 5 EVALUATION

We implemented and compared the following sketches:

- **Ours**: a sparse sketch for which both the values and the positions of the non-zero entries have been optimized
- **GD only**: a sparse sketch with optimized values and random positions for non-zero entries
- **Random**: classical random CountSketch

We also consider two "naïvely" learned sketches for LRA, which are computed on just one sample from the training set.

- **Exact SVD**: sketch as $AV_m$, where $V_m$ contains the top $m$ right singular vectors of random sample $A_i \in \mathcal{A}_{\mathrm{train}}$
- **Column sampling**: sketch as $AR$, where $R$ is computed from a randomly sampled $A_i \in \mathcal{A}_{\mathrm{train}}$. Each column of $R$ contains one entry. The location of this entry is sampled according to the squared column norms of $A_i$; the value of this entry is the inverse of the selected norm.

**Data sets.** We used high-dimensional data sets of *three different types* (image, text, and graph) to test the performance and versatility of our method. Note that the regression datasets are formed from the LRA/$k$-means datasets by splitting each matrix into two parts.

Table 5.1: Data set descriptions

| Name | Description | $A$ **shape** (for LRA, $k$-means) | $(A, B)$ **shapes** (for MRR) | $N_{train}$ | $N_{test}$ |
|---|---|---|---|---|---|
| Friends | Frames from a scene in the TV show *Friends*[2] | $5760 \times 1080$ | $(5760 \times 1079), (5760 \times 1)$ | 400 | 100 |
| Hyper | Hyperspectral images depicting outdoor scenes (Imamoglu et al., 2018) | $1024 \times 768$ | $(1024 \times 767), (1024 \times 1)$ | 400 | 100 |
| Logo | Frames from video of logo being painted[3] | $5760 \times 1080$ | $(5760 \times 1079), (5760 \times 1)$ | 400 | 100 |
| Yelp | tf-idf (Ramos, 2003) of restaurant reviews, grouped by restaurant[4] | $7000 \times 800$ | $(7000 \times 640), (7000 \times 160)$ | 260 | 65 |
| Graph | Graph adjacency matrices of social circles on Google+ (Leskovec & McAuley, 2012) | $1052 \times 1052$ | $(1052 \times 842), (1052 \times 210)$ | 147 | 37 |

---

[2] http://youtu.be/xmLZsEfXEgE
[3] http://youtu.be/L5HQoFIaT4I
[4] https://www.yelp.com/dataset

**Evaluation metric.**   To evaluate the quality of a sketch $S$, we compute the difference between the objective value of the sketched solution and the optimal solution (with no sketching). That is, the values in this section's tables denote

$$\Delta_S = \mathcal{L}_A(G(S, \mathcal{A}_{\text{test}})) - \mathcal{L}_A^*$$

averaged over 10 trials.

**Analysis of results.**   For least-squares regression and LRA, our method is the best. It is significantly better than random, obtaining improvements of around 31% and 70% for MRR and LRA respectively compared to a classical sparse sketch. For $k$-means, "Column Sampling" and "Exact SVD" dominated on several data sets. However, we note that our method was always a close second and also, more importantly, the values for $k$-means were only trivially different ($< 1\%$) between methods.

**Ablation of greedy stage**   We find empirically that including the greedy optimization stage is always better than not including it (compare the "Ours" vs. "SGD only" methods). For regression, LRA, and $k$-means, including this step offers around 12%, 20%, and $< 1\%$ improvement respectively. We should note that for $k$-means, the values generally do not differ much between methods.

Table 5.2: Average errors for least-squares regression

| Parameters | Algorithm | Datasets | | | | |
|---|---|---|---|---|---|---|
| $m$ | | Friends | Hyper | Logo | Yelp | Graph |
| 20 | Ours | **0.4360** ±5.6e−02 | **0.6242** ±9.7e−03 | **0.1942** ±1.2e−02 | **0.2734** ±1.8e−03 | **58.6568** ±6.6e−02 |
| | SGD only | 0.5442 ±9.0e−03 | 0.6790 ±1.2e−02 | 0.2547 ±3.7e−03 | 0.2781 ±6.3e−04 | 58.7231 ±8.7e−02 |
| | Random | 0.7792 ±2.5e−02 | 0.8226 ±4.4e−02 | 0.3773 ±1.5e−02 | 0.3173 ±1.3e−03 | 59.5535 ±1.2e−01 |
| 40 | Ours | **0.2485** ±1.1e−02 | **0.5380** ±8.3e−03 | **0.1607** ±5.7e−03 | **0.2639** ±4.1e−04 | **55.5267** ±8.0e−02 |
| | SGD only | 0.3688 ±9.0e−03 | 0.5718 ±1.2e−02 | 0.1858 ±2.8e−03 | 0.2701 ±7.2e−04 | 55.6120 ±6.5e−02 |
| | Random | 0.5213 ±1.6e−02 | 0.6414 ±1.9e−02 | 0.2389 ±6.9e−03 | 0.3114 ±2.6e−03 | 55.7896 ±6.7e−02 |

Table 5.3: Average error for LRA

| Parameters | Algorithm | Datasets | | | | |
|---|---|---|---|---|---|---|
| rank $k$, m | | Friends | Hyper | Logo | Yelp | Graph |
| (20, 40) | Ours | **0.8998** ±2.7e−02 | **2.4977** ±1.8e−01 | **0.5009** ±2.2e−02 | **0.1302** ±3.5e−04 | **30.0969** ±1.3e−01 |
| | SGD only | 1.0484 ±1.3e−02 | 3.7648 ±4.2e−02 | 0.6879 ±8.8e−03 | 0.1316 ±8.8e−04 | 30.5237 ±1.5e−01 |
| | Random | 4.0730 ±1.7e−01 | 6.3445 ±1.8e−01 | 2.3721 ±8.3e−02 | 0.1661 ±1.4e−03 | 33.0651 ±3.4e−01 |
| | Exact SVD | 1.5774 ±1.1e−01 | 3.4406 ±8.7e−01 | 0.7470 ±1.0e−01 | 0.1594 ±3.7e−03 | 31.0617 ±3.4e−01 |
| | Column sampling | 5.9837 ±6.6e−01 | 9.7126 ±8.2e−01 | 4.2008 ±6.0e−01 | 0.1881 ±3.2e−03 | 43.9920 ±5.6e−01 |
| (30, 60) | Ours | **0.7945** ±1.5e−02 | **2.4920** ±2.6e−01 | **0.4929** ±2.3e−02 | **0.1128** ±3.2e−04 | **30.6163** ±1.8e−01 |
| | SGD only | 1.0772 ±1.2e−02 | 3.7488 ±1.8e−02 | 0.7348 ±7.1e−03 | 0.1137 ±5.9e−04 | 30.9279 ±2.0e−01 |
| | Random | 2.6836 ±7.4e−01 | 5.3904 ±7.7e−02 | 1.6428 ±4.3e−02 | 0.1463 ±2.3e−03 | 32.7905 ±2.1e−01 |
| | Exact SVD | 1.1678 ±5.2e−02 | 3.0648 ±8.5e−01 | 0.5958 ±7.2e−02 | 0.1326 ±2.3e−03 | 32.0861 ±5.9e−01 |
| | Column sampling | 4.1899 ±2.9e−01 | 8.2314 ±4.7e−01 | 3.2296 ±3.1e−01 | 0.1581 ±3.9e−03 | 44.2672 ±5.2e−01 |

Table 5.4: Average errors for $k$-means clustering

| Algorithm | Parameters | Data sets | | | | |
|---|---|---|---|---|---|---|
| # clusters, $m$, rank $k$ | | Friends | Hyper | Logo | Yelp | Graph |
| (20, 40, 20) | Ours | 0.2934 ±5.8e−05 | **0.6110** ±1.7e−04 | 0.1251 ±1.2e−04 | **2.4755** ±7.2e−04 | 2.9797 ±1.9e−03 |
| | SGD only | 0.2935 ±1.4e−04 | 0.6117 ±2.0e−04 | 0.1253 ±5.8e−05 | 2.4756 ±7.6e−04 | 2.9810 ±9.9e−04 |
| | Random | 0.2933 ±1.5e−04 | 0.6122 ±2.6e−04 | 0.1254 ±1.3e−04 | 2.4759 ±7.7e−04 | 2.9834 ±1.8e−03 |
| | Exact SVD | 0.2934 ±1.2e−04 | 0.6113 ±2.5e−04 | 0.1253 ±2.5e−04 | 2.4797 ±9.0e−04 | **2.9779** ±1.4e−03 |
| | Column sampling | **0.2932** ±1.7e−04 | 0.6129 ±3.7e−04 | **0.1250** ±2.7e−04 | 2.4922 ±3.8e−03 | 3.0137 ±2.9e−03 |
| (30, 60, 30) | Ours | 0.2604 ±7.6e−05 | 0.5448 ±1.4e−04 | 0.1062 ±7.7e−05 | **2.4475** ±6.8e−04 | 2.8765 ±1.1e−03 |
| | SGD only | 0.2605 ±4.9e−05 | 0.5452 ±1.6e−04 | 0.1063 ±1.2e−04 | 2.4476 ±7.1e−04 | 2.8777 ±1.2e−03 |
| | Random | 0.2604 ±9.3e−05 | 0.5453 ±1.6e−04 | 0.1062 ±4.6e−05 | 2.4477 ±7.4e−04 | 2.8791 ±1.4e−03 |
| | Exact SVD | 0.2605 ±1.8e−04 | **0.5446** ±2.6e−04 | 0.1063 ±4.4e−05 | 2.4532 ±4.6e−04 | **2.8739** ±1.7e−03 |
| | Column sampling | **0.2602** ±1.2e−04 | 0.5459 ±3.6e−04 | **0.1059** ±1.1e−04 | 2.4665 ±1.1e−03 | 2.9002 ±2.2e−03 |

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

## A  APPENDIX

### CORRESPONDING TO SECTION 4.1: LEAST-SQUARES REGRESSION

**Definition A.1 (Subspace Embedding).** *Given an $n \times d$ matrix $A$, a matrix $S$ is an* affine $\epsilon$-embedding *for (the column space of) $A$ if for $x \in \mathbb{R}^d$, $\|SAx\|_2^2 = (1 \pm \epsilon) \|Ax\|_2^2$.*

The following result is shown in Clarkson & Woodruff (2017) and sharpened with Nelson & Nguyên (2013); Meng & Mahoney (2013).

**Lemma A.2.** *Given matrices $A, B$ with $n$ rows, a sparse embedding matrix (i.e., random CS) with $O(\text{rank}(A)^2/\epsilon^2)$ rows is an affine $\epsilon$-embedding matrix with constant probability. Moreover, the matrix product $SA$ can be computed in $O(\text{nnz}(A))$ time, where $\text{nnz}(A)$ denotes the number of non-zero entries of matrix $A$.*

**Lemma A.3 (Clarkson & Woodruff (2017); Lemma 40).** *Let $A$ be an $n \times d$ matrix and let $S \in \mathbb{R}^{O(\frac{1}{\epsilon^2}) \times n}$ be a randomly chosen sparse embedding matrix (i.e., random CS). Then with constant probability, $\|SA\|_F^2 = (1 \pm \epsilon) \|A\|_F^2$.*

**Lemma A.4 (Sarlos (2006); Clarkson & Woodruff (2017)).** *Given classical CountSketch $S \in \mathbb{R}^{m \times n}$, SKETCH-REGRESSION$(A, B, S)$ returns a $(1 + \epsilon)$-approximation in time $O(\text{nnz}(A) + \text{nnz}(B) + dd'm^2 + \min(d^2m, dm^2))$.*

*Proof:* Since $S$ is an affine $\epsilon$-embedding matrix of $A, B$, then

$$\|SAX - SB\|_F^2 = (1 \pm \epsilon) \|AX - B\|_F^2$$

Next, by the normal equations, $(SA)^+(SB)$ is a minimizer of $\min_X \|SAX - SB\|_F^2$ and is a $(1 + 3\epsilon)$-approximate solution of $\min_X \|AX - B\|_F^2$.

To bound the runtime, note that since $S$ is a CountSketch we can compute $SA$ and $SB$ in time $O(\text{nnz}(A) + \text{nnz}(B))$ and reduce the problem to an instance of multiple-response regression with $m$ rows. Then, we can solve the reduced size problem in time $O(dd'm^2 + \min(d^2m, dm^2))$: $O(\min(dm^2, d^2m))$ to compute $(SA)^+$ and $O(dd'm^2)$ to compute $(SA)^+(SB)$. □

*Proof of* **Theorem 4.1.** By Lemma A.2, a random CS $S_O$ with $O(\frac{d^2}{\epsilon^2})$ rows is an affine $\epsilon$-embedding matrix of $A, B$ with constant probability.

Next, let $X_L$ and $X_O$ be respectively the solutions returned by SKETCH-REGRESSION$(A, B, S_L)$ and SKETCH-REGRESSION$(A, B, S_O)$. By Lemma A.4, with constant probability,

$$\|AX_O - B\|_F^2 \le (1 + 3\epsilon) \min_X \|AX - B\|_F^2 .$$

Together with the assumption that the solution constructed from $S_L$, which is denoted as $X_L$, over $\mathcal{A}_{\text{train}}$ is a $\gamma$-approximate solution,

$$\min(\|AX_L - B\|_F^2, \|AX_O - B\|_F^2) \leq (1 + \min(3\epsilon, \gamma)) \min_X \|AX - B\|_F^2. \tag{A.1}$$

Hence, it only remains to compute the minimum of $\|AX_L - B\|_F^2$ and $\|AX_O - B\|_F^2$ efficiently. Note that it takes $\Omega(n \cdot d \cdot d')$ to compute these values *exactly*. However, for our purpose it suffices to compare $(1 + \beta)$-estimates of these values and return the minimum of the estimates. To achieve this, we use two applications of Lemma A.3 with $R^\top \in \mathbb{R}^{O(\frac{1}{\beta^2}) \times d'}$, $S \in \mathbb{R}^{O(\frac{1}{\beta^2}) \times n}$. For any $X'$ (in particular, both $X_O$ and $X_L$), with constant probability,

$$\|S(AX' - B)R\|_F^2 = \|R^\top(AX' - B)^\top S^\top\|_F^2 = (1 \pm \beta) \|(AX' - B)\|_F^2 \tag{A.2}$$

Let $\Gamma_L$ and $\Gamma_O$ respectively denote $AX_L - B$ and $AX_O - B$ and let $\Gamma_M = \arg\min(\|S\Gamma_L R\|_F, \|S\Gamma_O R\|_F)$. By Eq. (A.2) and union bound over of $X_O$ and $X_L$, with constant probability,

$$
\begin{aligned}
\|\Gamma_M\| &\leq (1 + O(\beta)) \cdot \|S\Gamma_M R\|_F^2 && \triangleright \text{ by Lemma A.3} \\
&\leq (1 + O(\beta)) \cdot \min(\|\Gamma_O\|_F^2, \|\Gamma_L\|_F^2) && \triangleright \text{ by Lemma A.3} \\
&\leq (1 + O(\beta + \min(\epsilon, \beta))) \|AX - B\|_F^2 && \triangleright \text{ by Eq. (A.1)}
\end{aligned}
$$

**Runtime Analysis.** By Lemma A.4, $X_O$ an $X_L$ can be computed in $O(\text{nnz}(A) + \text{nnz}(B) + d^5 d' \epsilon^{-4})$ time. Next, the time to compute $\Delta_L$ and $\Delta_O$ is

$$O(\text{nnz}(A) + \text{nnz}(XR) + \text{nnz}(B) + \frac{d}{\beta^4}) = O(\text{nnz}(A) + \text{nnz}(B) + dd' + \frac{d}{\beta^4}),$$

where $X$ is either $X_O$ or $X_L$ and we use that fact that $\text{nnz}(X) \leq d \cdot d'$ (i.e., the total number of cells in $X$).

Thus, the total runtime of Algorithm 3 is $O(\text{nnz}(A) + \text{nnz}(B) + d^5 d' \epsilon^{-4} + d\beta^{-4})$. $\qquad\square$

**Theorem A.5 (Least Squares Regression).** *Suppose there exists a learned, sparse, subspace embedding matrix $S_L$ computed over $\mathcal{A}_{\text{train}}$ with $\text{poly}(\frac{d}{\epsilon})$ rows that attains a $\beta$-approximation over $\mathcal{A}_{\text{test}}$. Then, there exists an algorithm that runs in time $O(\text{nnz}(A) + \text{poly}(d/\epsilon))$ that outputs a $(1 + \min(\beta, \epsilon))$-approximation to the least squares regression problem.*

*Proof:* Note that to solve $\min_{x \in \mathbb{R}^d} \|Ax - b\|_2$, given a sparse embedding matrix $S \in \mathbb{R}^{m \times n}$ for the columns of $A$ together with the vector $b$, the problem can be solved within $(1 + \epsilon)$-approximation in time $\text{nnz}(A) + \text{poly}(d/\epsilon)$ (e.g., see Theorem 2.14 Woodruff (2014)).

The proof outline is similar to the proof of Theorem 4.1. First, we compute solutions $x_L, x_O \in \mathbb{R}^d$ to the given instance using respectively a learned sketching matrix $S_L$ and a learning-free sketching algorithm. Then, we compare $\|Ax_L - b\|_2$ and $\|Ax_O - b\|_2$ and report the better. Note that since $x_L, x_O$ are vectors, unlike the case of MRR, the naive comparison (i.e., without applying any sketching matrices) takes $\text{nnz}(A)$. Hence, the algorithm runs in time $\text{nnz}(A) + \text{poly}(d/\epsilon)$ and returns a $(1 + \min(\epsilon, \beta))$-approximate solution of the least squares regression problem.

Moreover, we can employ the best known sketching techniques for the least squares regression problem and achieve the dependence $d/\epsilon^2$ in the running time (e.g., see Section 2.5 of Woodruff (2014)). $\qquad\square$

## A.1 Corresponding to Section 4.2: Low-rank approximation

**Lemma A.6.** *Suppose that $S \in \mathbb{R}^{m_S \times n}$ and $R \in \mathbb{R}^{m_R \times d}$ are sparse affine $\epsilon$-embedding matrices for $(A^\top, A)$ and $((SA)^\top, A^\top)$. Then,*

$$\min_{\text{rank-}k\ X} \|AR^\top XSA - A\|_F^2 \leq (1 + \epsilon) \|A_k - A\|_F^2$$

*Proof:* Consider the following multiple-response regression problem:

$$\min_{\text{rank-}k\ X} \|A_k X - A\|_F^2. \tag{A.3}$$

Note that since $X = I_k$ is a feasible solution to Eq. (A.3), $\min_{\text{rank-}k\ X} \|A_k X - A\|_F^2 = \|A_k - A\|_F^2$. Let $S \in \mathbb{R}^{m_S \times n}$ be a sketching matrix that satisfies the condition of Lemma A.9 for $A := A_k$ and $B := A$. By the normal equations, the rank-$k$ minimizer of $\|SA_k X - SA\|_F^2$ is $(SA_k)^+ SA$. Hence,

$$\left\| A_k (SA_k)^+ SA - A \right\|_F^2 \leq (1 + \epsilon) \|A_k - A\|_F^2, \tag{A.4}$$

which in particular shows that a $(1 + \epsilon)$ rank-$k$ approximation of $A$ exists in the row space of $SA$. In other words,

$$\min_{\text{rank-}k\ X} \|XSA - A\|_F^2 \leq (1 + \epsilon) \|A_k - A\|_F^2. \tag{A.5}$$

Next, let $R \in \mathbb{R}^{m_R \times d}$ be a sketching matrix that satisfies the condition of Lemma A.9 for $A := (SA)^\top$ and $B := A^\top$. Let $Y$ denote the rank-$k$ minimizer of $\left\| R(SA)^\top X^\top - RA^\top \right\|_F^2$. Hence,

$$\begin{aligned}
\left\| (SA)^\top Y^\top - A^\top \right\|_F^2 &\leq (1 + \epsilon) \min_{\text{rank-}k\ X} \|XSA - A\|_F^2 && \triangleright \text{ Lemma A.9} \\
&\leq (1 + O(\epsilon)) \|A_k - A\|_F^2 && \triangleright \text{ Eq. (A.5)} \tag{A.6}
\end{aligned}$$

Note that by the normal equations, again $\text{rowsp}(Y^\top) \subseteq \text{rowsp}(RA^\top)$ and we can write $Y = AR^\top Z$ where $\text{rank}(Z) = k$. Thus,

$$\begin{aligned}
\min_{\text{rank-}k\ X} \left\| AR^\top XSA - A \right\|_F^2 &\leq \left\| AR^\top ZSA - A \right\|_F^2 \\
&= \left\| (SA)^\top Y^\top - A^\top \right\|_F^2 && \triangleright\ Y = AR^\top Z \\
&\leq (1 + O(\epsilon)) \|A_k - A\|_F^2 && \triangleright \text{ Eq. (A.6)} \qquad \square
\end{aligned}$$

**Lemma A.7 (Avron et al. (2016); Lemma 27).** *For $C \in \mathbb{R}^{p \times m'}, D \in \mathbb{R}^{m \times p'}, G \in \mathbb{R}^{p \times p'}$, the following problem*

$$\min_{\text{rank-}k\ Z} \|CZD - G\|_F^2 \tag{A.7}$$

*can be solved in $O(pm'r_C + p'mr_D + pp'(r_D + r_C))$ time, where $r_C = \text{rank}(C) \leq \min\{m', p\}$ and $r_D = \text{rank}(D) \leq \min\{m, p'\}$.*

*Proof:* Let $U_C$ and $U_D^\top$ be orthogonal bases for $\text{colsp}(C)$ and $\text{rowsp}(D)$, respectively, so that for each $Z$, $CZD = U_C Z' U_D^\top$ for some $Z'$. Let $P_C$ and $P_D$ be the projection matrices onto the subspaces spanned by the rows of $C^\top$ and $D^\top$, respectively: $P_C = U_C U_C^\top$ and $P_D = U_D U_D^\top$. Then by the Pythagorean theorem,

$$\begin{aligned}
\|CZD - G\|_F^2 &= \left\| P_C U_C Z' U_D^\top P_D - G \right\|_F^2 \\
&= \left\| P_C U_C Z' U_D^\top P_D - P_C G P_D \right\|_F^2 + \|P_C G(I - P_D)\|_F^2 + \|(I - P_C)G\|_F^2,
\end{aligned}$$

where the first equality holds since $P_C U_C = U_C$ and $U_D^\top P_D = U_D^\top$ and the second equality follows from the Pythagorean theorem. Hence,

$$\arg\min_{\text{rank-}k\ Z} \|CZD - G\|_F^2 = \arg\min_{\text{rank-}k\ Z} \left\| P_C U_C Z U_D^\top P_D - P_C G P_D \right\|_F^2.$$

Moreover,

$$\left\| P_C U_C Z U_D^\top P_D - P_C G P_D \right\|_F^2 = \left\| U_C Z U_D^\top - U_C U_C^\top G U_D U_D^\top \right\|_F^2 = \left\| Z - U_C^\top G U_D \right\|_F^2,$$

where the first equality holds since $U_C^\top U_C = I$ and $U_D^\top U_D = I$, and the second equality holds since $U_C$ and $U_D^\top$ are orthonormal. Hence,

$$\arg\min_{\text{rank-}k\ Z} \|CZD - G\|_F^2 = \arg\min_{\text{rank-}k\ Z} \left\| Z - U_C^\top G U_D \right\|_F^2.$$

Next, we can find $Z = [U_C^\top G U_D]_k$ by computing the SVD of $U_C^\top G U_D$ in form of $Z_L Z_R$ where $Z_L \in \mathbb{R}^{m' \times k}$ and $Z_R \in \mathbb{R}^{k \times m}$.

**Runtime analysis.** We can compute $U_C$ and $U_D$ by the Gram-Schmidt process in time $O(pm'r_C + p'mr_D)$, and $U_C^\top G U_D$ in time $O(\min\{r_C p'(p + r_D), r_D p(p' + r_C)\})$. Finally, the time to compute $Z$ (i.e., an SVD computation of $U_C^\top G U_D$) is $O(r_C r_D \cdot \min\{r_C, r_D\})$. Since $r_C \leq \min\{p, m'\}$ and $r_D \leq \min\{p', m\}$, the total runtime to minimize $Z$ in Eq. (A.7) is $O(pm'r_C + p'mr_D + pp'(r_C + r_D))$. $\qquad\square$

**Lemma A.8.** *Let $S \in \mathbb{R}^{\mathrm{poly}(k/\epsilon) \times d}$, $R \in \mathbb{R}^{\mathrm{poly}(k/\epsilon) \times d}$ be CS matrices such that*

$$\min_{\mathrm{rank}\text{-}k\ X} \left\| AR^\top XSA - A \right\|_F^2 \leq (1 + \gamma) \left\| A_k - A \right\|_F^2. \tag{A.8}$$

*Moreover, let $S_2 \in \mathbb{R}^{\frac{k^2}{\beta^2} \times n}$, and $R_2 \in \mathbb{R}^{\frac{k^2}{\beta^2} \times d}$ be CS matrices. Then, Algorithm 5 runs in $O(\mathrm{nnz}(A) + (n + d)\,\mathrm{poly}(k/\epsilon))$ time and with constant probability gives a $(1 + O(\beta + \gamma))$-approximate $\mathrm{rank}$-$k$ approximation of $A$.*

*Proof:* The algorithm first computes $C = S_2 A R^\top, D = SAR_2^\top, G = S_2 A R_2^\top$ which can be done in time $O(\mathrm{nnz}(A))$. As an example, we bound the time to compute $C = S_2 A R$. Note that since $S_2$ is a CS, $S_2 A$ can be computed in $O(\mathrm{nnz}(A))$ time and the number of non-zero entries in the resulting matrix is at most $\mathrm{nnz}(A)$. Hence, since $R$ is a CS as well, $C$ can be computed in time $O(\mathrm{nnz}(A) + \mathrm{nnz}(S_2 A)) = O(\mathrm{nnz}(A))$. Then, it takes an extra $\mathrm{poly}(k/\epsilon) \cdot \frac{k^2}{\beta^2}$ time to store $C, D$ and $G$ in matrix form. Next, as we showed in Lemma A.7, the time to compute $Z$ in Algorithm 5 is $O(\frac{k^4}{\beta^4} \cdot \mathrm{poly}(k/\epsilon))$. Finally, it takes $(n + d)\,\mathrm{poly}(k/\epsilon)$ time to compute $Q = AR^\top Z_L$ and $P = Z_R SA$ and return the solution in the form of $P_{n \times k} Q_{k \times d}$. Hence, the total runtime is $O(\mathrm{nnz}(A) + (n + d)\,\mathrm{poly}(k/\epsilon) + \frac{k^4}{\beta^4} \cdot \mathrm{poly}(k/\epsilon))$.

The approximation guarantee follows from Eq. (A.8) and the fact that $S_2$ and $R_2$ are respectively affine $\beta$-embedding matrices of $AR^\top$ and $SA$ (see Lemma A.2). $\qquad\square$

**Lemma A.9 (Avron et al. (2016); Lemma 25).** *Suppose that $A \in \mathbb{R}^{n \times d}$ and $B \in \mathbb{R}^{n \times d'}$. Moreover, let $S$ be an oblivious sparse affine $\epsilon$-embedding matrix (i.e., a random CS) with $(\mathrm{rank}(A)^2/\epsilon^2)$ rows. Then with constant probability, $\tilde{X} = \arg\min_{\mathrm{rank}\text{-}k\ X} \|SAX - SB\|_F^2$, satisfies*

$$\left\| A\tilde{X} - B \right\|_F^2 \leq (1 + \epsilon) \min_{\mathrm{rank}\text{-}k\ X} \|AX - B\|_F^2.$$

*In other words, in $O(\mathrm{nnz}(A) + \mathrm{nnz}(B)) + (d + d')(\mathrm{rank}(A)^2/\epsilon^2)$ time, we can reduce the problem to a smaller (multi-response regression) problem with $(\mathrm{rank}(A)^2/\epsilon^2)$ rows whose optimal solution is a $(1 + \epsilon)$-approximate solution to the original problem.*

*Proof of* **Theorem 4.3.** Let $S_O$ and $R_O$ be CountSketch matrices of size $\mathrm{poly}(k/\epsilon) \times n$ and $\mathrm{poly}(k/\epsilon) \times d$. Note that since $\mathrm{rank}(A_k) = k$ and $\mathrm{rank}((S_O A)^\top) \leq \mathrm{poly}(k/\epsilon)$, $S_O$ and $R_O$ are respectively affine $\epsilon$-embedding matrices of $(A_k, A)$ and $((S_O A)^\top, A^\top)$. Then, by an application of Lemma A.6

$$\min_{\mathrm{rank}\text{-}k\ X} \left\| AR_O^\top XS_O A - A \right\|_F^2 \leq (1 + O(\epsilon)) \left\| A_k - A \right\|_F^2 \tag{A.9}$$

Similarly, by the assumption that $S_L$ and $R_L$ finds a $(1 + \gamma)$-approximate solution of LRA over matrices $A \in \mathcal{A}_{\mathrm{test}}$, then

$$\min_{\mathrm{rank}\text{-}k\ X} \left\| AR_L^\top XS_L A - A \right\|_F^2 \leq (1 + O(\gamma)) \left\| A_k - A \right\|_F^2 \tag{A.10}$$

Next, let $(P_L, Q_L)$ and $(P_O, Q_O)$ be respectively the $\mathrm{rank}$-$k$ approximations of $A$ in factored form using $(S_L, R_L)$ and $(S_O, R_O)$ (see Algorithm 5). Then, Eq. (A.10) together with Eq. (A.9) implies that

$$\min(\|P_L Q_L - A\|_F^2, \|P_O Q_O - A\|_F^2) = (1 + O(\min(\epsilon, \gamma))) \|A_k - A\|_F^2 \tag{A.11}$$

Hence, it only remains to compute the minimum of $\|P_L Q_L - A\|_F^2$ and $\|P_O Q_O - A\|_F^2$ efficiently and we proceed similarly to the proof of Theorem 4.1. We use two applications of Lemma A.3 with $R^\top \in \mathbb{R}^{O(\frac{1}{\beta^2}) \times d}, S \in \mathbb{R}^{O(\frac{1}{\beta^2}) \times n}$. Let $\Gamma_L = P_L Q_L - A$, $\Gamma_O = P_O Q_O - A$ and $\Gamma_M = \arg\min(\|S\Gamma_L R\|_F, \|S\Gamma_O R\|_F)$. Hence,

$$\begin{aligned}
\|\Gamma_M\|_F^2 &\leq (1 + O(\beta)) \|S\Gamma_M R\|_F^2 && \triangleright \text{ by Lemma A.3} \\
&\leq (1 + O(\beta)) \cdot \min(\|\Gamma_L\|_F^2, \|\Gamma_O\|_F^2) && \triangleright \text{ by Lemma A.3} \\
&\leq (1 + O(\beta + \min(\epsilon, \gamma))) \|A_k - A\|_F^2 && \triangleright \text{ by Eq. (A.11)}
\end{aligned}$$

**Runtime analysis.** By Lemma A.8, Algorithm 5 computes $P_L, Q_L$ and $P_O, Q_O$ in $O(\mathrm{nnz}(A) + (n+d)\,\mathrm{poly}(\frac{k}{\epsilon}) + \frac{k^4}{\beta^4} \cdot \mathrm{poly}(\frac{k}{\epsilon}))$.

Next, it takes $O(\mathrm{nnz}(A) + (n+d) \cdot k + \frac{k}{\beta^4})$ to compute $\Delta_L$ and $\Delta_O$. As an example, we bound the amount of time required to compute $SP_LQ_LR - SAR$ corresponding to $\Delta_L$. Since $S$ and $R$ are sparse sketching matrices, $SP_L, Q_LR$ and $SAR$ can be computed in $\mathrm{nnz}(SP_L) + \mathrm{nnz}(Q_LR) + \mathrm{nnz}(A)$. Since $SP_L$ and $Q_LR$ are respectively of size $\frac{1}{\beta^2} \times k$ and $k \times \frac{1}{\beta^2}$, in total it takes $O(\mathrm{nnz}(A) + \frac{k}{\beta^2})$ to compute these three matrices. Then, we can compute $SP_LQ_LR$ and $\|SP_LQ_LRSAR\|_F$ in time $O(\frac{k}{\beta^4})$.

Hence, the total runtime of Algorithm 3 for LRA is $O(\mathrm{nnz}(A) + (n+d) \cdot \mathrm{poly}(\frac{k}{\epsilon}) + \frac{k^4}{\beta^4} \cdot \mathrm{poly}(\frac{k}{\epsilon}))$.□

## A.2 CORRESPONDING TO SECTION 4.3: $k$-MEANS

We restate notation and the main result below for ease of reference.

**Notation.** We define $A_m$ as the optimal rank-$m$ approximation of $A$ formed by truncated SVD: $A_m = U_m \Sigma_m V_m^\top$.

Given a matrix $U$ with orthogonal columns, let $\pi_U(A) = AUU^\top$, which is the projection of the rows of $A$ onto $col(U)$. Let $\mathcal{C}_U$ be the optimal $k$-means partition of $\pi_U(A)$. Further, we let $\mu_{C_{U,i}}$ denote the $i$-th cluster's center in the optimal $k$-means clustering on $\pi_U(A)$.

We denote $\mathrm{dist}^2(A, \mu)$ as the $k$-means loss given cluster centers (and their corresponding partition):

$$\mathrm{dist}^2(A, \mu) = \sum_{i \in [k]} \sum_{j \in C_i} \|A_j - \mu_i\|_2^2$$

Likewise, $\mathrm{cost}(\mathcal{C})$ is the $k$-means loss given a partition:

$$\mathrm{cost}(\mathcal{C}) = \sum_{i \in [k]} \min_{\mu_i} \sum_{j \in C_i} \|A_j - \mu_i\|_2^2$$

**Definition A.10 (Projection-cost preserving sketch).** $\tilde{A}$ *is a projection-cost preserving sketch of* $A$ *if for any low rank projection matrix* $P$ *and* $c$ *not dependent on* $P$:

$$(1 - \epsilon) \|A - PA\|_F^2 \leq \left\|\tilde{A} - P\tilde{A}\right\|_F^2 + c \leq (1 + \epsilon) \|A - PA\|_F^2$$

**Theorem (4.6: Sketch monotonicity property for $k$-means).** *Assume we have* $A \in \mathbb{R}^{n \times d}$. *We also have random CountSketch* $S \in \mathbb{R}^{O(\mathrm{poly}(k/\epsilon)) \times n}$ *and define* $U \in \mathbb{R}^{d \times O(\mathrm{poly}(k/\epsilon))}$ *with orthogonal columns such that* $\mathrm{colsp}(U) = \mathrm{rowsp}(SA)$. *Then, any extension of* $S$ *to* $S'$ *(for example, concatenation with a learned CountSketch* $S_L$*) yields a better approximate* $k$-means *approximation. Specifically, define* $W$ *with orthogonal columns such that* $col(W) = row(S'A)$. *Let* $\mathcal{C}^*$ *denote the optimal partition of* $A$, $\mathcal{C}_U$ *denote the optimal partition of* $\pi_U(A)$, *and* $\mathcal{C}_W$ *denote the optimal partition of* $\pi_W(A)$. *Then*

$$\mathrm{cost}(\mathcal{C}_W) \leq (1 + O(\epsilon))\mathrm{cost}(\mathcal{C}_U) \leq (1 + O(\epsilon))\mathrm{cost}(\mathcal{C}^*)$$

*Proof:*

$$\begin{aligned}
\mathrm{cost}(\mathcal{C}_W) &= \sum_{i \in [k]} \min_{\mu_i} \sum_{j \in C_{W,i}} \|A_j - \mu_i\|^2 \\
&\leq \sum_{i \in [k]} \sum_{j \in C_{W,i}} \left\|A_j - \mu_{C_{W,i}}\right\|^2 \\
&= \sum_{i \in [k]} \sum_{j \in C_{W,i}} \|A_j - \pi_W(A_j)\|^2 + \left\|\pi_W(A_j) - \mu_{C_{W,i}}\right\|^2 \quad\quad (A.12)
\end{aligned}$$

$$\leq \sum_{i \in [k]} \sum_{j \in C_{U,i}} \|A_j - \pi_W(A_j)\|^2 + \left\|\pi_W(A_j) - \mu_{C_{U,i}}\right\|^2 \tag{A.13}$$

$$= \sum_{i \in [k]} \sum_{j \in C_{U,i}} \left\|A_j - \mu_{C_{U,i}}\right\|^2 \tag{A.14}$$

$$\leq (1 + \epsilon) \sum_{i \in k} \min_{\mu_i} \sum_{j \in C_{U,i}} \|A_j - \mu_i\|^2 \tag{A.15}$$

$$= (1 + \epsilon)\text{cost}(\mathcal{C}_U)$$
$$\leq (1 + O(\epsilon))\text{cost}(\mathcal{C}^*) \qquad \qquad \square$$

(A.12): $\mu_{\mathcal{C}_W} \in \text{colsp}(W)$ so we can apply the Pythagorean Theorem.

(A.13): $(\mu_{\mathcal{C}_W}, \mathcal{C}_W)$ is an optimal $k$-means clustering of the projected points $\pi_W(A)$.

(A.14): $\mu_{\mathcal{C}_U} \in \text{colsp}(U) \subset \text{colsp}(W)$, so we can apply the Pythagorean Theorem.

(A.15): We apply Corollary A.14.

**Remark A.11.** Our result shows that the "sketch monotonicity" property holds for sketching matrices that provide strong coresets for $k$-means clustering. Besides strong coresets, an alternate approach to showing that the clustering objective is approximately preserved on sketched inputs is to show a weaker property: the clustering cost is preserved for *all possible partitions* of the points into $k$ groups Makarychev et al. (2019). While the dimension reduction mappings satisfying strong coresets require $\text{poly}(k/\epsilon)$ dimensions, Makarychev et al. (2019) shows that $O(\log k/\epsilon^2)$ dimensions suffice to satisfy this "partition" guarantee. An interesting question for further research is if the sketch monotonicity guarantee also applies to the construction of Makarychev et al. (2019).

**Corollary A.12.** *Assume we have $A \in \mathbb{R}^{n \times d}$, $j \in \mathbb{Z}^+$, $\epsilon > 0$. Define $m = \min(O(\text{poly}(j/\epsilon)), d)$. Let $\tilde{A}_m$ be a $(1 + \epsilon)$-approximation to $A_m$ of the form $\tilde{A}_m = AVV^\top$ where $SA = U\Sigma V^\top$ for CountSketch $S \in \mathbb{R}^{m \times n}$. Let $X \in \mathbb{R}^{d \times j}$ be a matrix whose columns are orthonormal, and let $Y \in \mathbb{R}^{d \times (d-j)}$ be a matrix with orthonormal columns that spans the orthogonal complement of $colsp(X)$. Then*

$$\left\|AXX^\top - \tilde{A}_m XX^\top\right\|_F^2 \leq \epsilon \cdot \|AY\|_F^2.$$

*Proof:*

$$\left\|AXX^\top - \tilde{A}_m XX^\top\right\|_F^2 = \left\|A(I - VV^\top)XX^\top\right\|_F^2$$

$$\leq j \left\|A(I - VV^\top)XX^\top\right\|_2^2 \tag{A.16}$$

$$\leq j \left\|A(I - VV^\top)\right\|_2^2 \tag{A.17}$$

$$\leq j \cdot O(\frac{\epsilon}{j}) \|A - A_j\|_F^2 \tag{A.18}$$

$$= O(\epsilon) \sum_{i=j}^{\min(n,d)} \sigma_i^2 \tag{A.19}$$

$$\leq O(\epsilon) \|AY\|_F^2 \tag{A.20}$$

$$\square$$

(A.16) Note that $\text{rank}(X) = j$, so $\text{rank}(A(I - VV^\top)XX^\top) = j$. We use this fact to bound the Frobenius norm by the operator norm.
(A.17) Using the fact that $XX^\top$ is a projection.
(A.18) Using Lemma 18 from Cohen et al. (2015), where $A_j$ is the optimal rank-$j$ approximation to $A$. We can apply this lemma because CountSketch is one of the eligible types of random projection matrices.
(A.19) Letting $\sigma_i$ be the singular values of $A$.
(A.20) $\sum_{i=j}^{\min(n,d)} \sigma_i^2 = \min_Y \|AY\|_F^2$ for $Y \in \mathbb{R}^{d \times (d-j)}$ with orthonormal columns.

**Theorem A.13.** *Assume we have* $A \in \mathbb{R}^{n \times d}$, $j \in \mathbb{Z}^+$, $\epsilon \in (0, 1]$. *Define* $m = \min(O(\text{poly}(j/\epsilon)), d)$. *Let* $\tilde{A}_m$ *be a* $(1 + \epsilon)$-*approximation to* $A_m$ *of the form* $\tilde{A}_m = AVV^\top$ *where* $SA = U\Sigma V^\top$ *for CountSketch* $S \in \mathbb{R}^{m \times n}$. *Then, for any non-empty set* $\mu$ *contained in a* $j$-*dimensional subspace, we have:*

$$\left| \text{dist}^2(\tilde{A}_m, \mu) + \left\| \tilde{A}_m - A \right\|_F^2 - \text{dist}^2(A, \mu) \right| \leq \epsilon \, \text{dist}^2(A, \mu)$$

*Proof:* We follow the proof of Theorem 22 in Feldman et al. (2013), but substitute different analyses in place of Corollaries 16 and 20. The result of Feldman et al. (2013) involves the *best* rank-$m$ approximation of $A$, $A_m$; we will show it for the *approximate* rank-$m$ approximation, $\tilde{A}_m$.

Define $X \in \mathbb{R}^{d \times j}$ with orthonormal columns such that $\text{colsp}(X) = \text{span}(\mu)$. Likewise, define $Y \in \mathbb{R}^{d \times (d-j)}$ with orthonormal columns such that $\text{colsp}(Y) = \text{span}(\mu)^\perp$. By the Pythagorean theorem:

$$\text{dist}^2(\tilde{A}_m, \mu) = \| \tilde{A}_m Y \|_F^2 + \text{dist}^2(\tilde{A}_m X X^T, \mu)$$

and

$$\text{dist}^2(A, \mu) = \| AY \|_F^2 + \text{dist}^2(AXX^T, \mu). \tag{A.21}$$

Hence,

$$\left| \left( \text{dist}^2(\tilde{A}_m, \mu) + \left\| A - \tilde{A}_m \right\|_F^2 \right) - \text{dist}^2(A, \mu) \right|$$

$$= \left| \| \tilde{A}_m Y \|_F^2 + \text{dist}^2(\tilde{A}_m X X^T, \mu) + \left\| A - \tilde{A}_m \right\|_F^2 - \left( \| AY \|_F^2 + \text{dist}^2(AXX^T, \mu) \right) \right|$$

$$\leq \left| \| \tilde{A}_m Y \|_F^2 + \| A - \tilde{A}_m \|_F^2 - \| AY \|_F^2 \right| + \left| \text{dist}^2(\tilde{A}_m X X^T, \mu) - \text{dist}^2(AXX^T, \mu) \right| \tag{A.22}$$

$$\leq \frac{\varepsilon^2}{8} \cdot \| AY \|_F^2 + \left| \text{dist}^2(\tilde{A}_m X X^T, \mu) - \text{dist}^2(AXX^T, \mu) \right| \tag{A.23}$$

$$\leq \frac{\varepsilon^2}{8} \cdot \text{dist}^2(A, \mu) + \left| \text{dist}^2(\tilde{A}_m X X^T, \mu) - \text{dist}^2(AXX^T, \mu) \right| \quad \triangleright \text{Used (A.21)} \tag{A.24}$$

(A.22) Triangle inequality.

(A.23) Take $\epsilon$ in Theorem 16 from Cohen et al. (2015) as $\epsilon^2/8$. This theorem implies that $\tilde{A}_m$ is a projection-cost preserving sketch with the $c$ term as $\left\| A - \tilde{A}_m \right\|_F^2$. Specifically, it says $AV$ is a project-cost preserving sketch, which means $\tilde{A}_m = AVV^\top$ is too: $V$ has orthonormal columns so $\| (I - P)AV \|_F^2 = \left\| (I - P)AVV^\top \right\|_F^2$.

By Corollary A.12,

$$\left\| \tilde{A}_m X X^T - AXX^T \right\|_F^2 \leq \frac{\epsilon^2}{8} \cdot \| AY \|_F^2.$$

Since $\mu \in \text{rowsp}(X)$, we have $\| AY \|_F^2 \leq \text{dist}^2(A, \mu)$. Using Corollary 21 from Feldman et al. (2013) while taking $\epsilon$ as $\epsilon/4$, $A$ as $\tilde{A}_m X X^T$, and $B$ as $AXX^T$ yields

$$| \text{dist}^2(\tilde{A}_m X X^T, \mu) - \text{dist}^2(AXX^T, \mu) | \leq \frac{\epsilon}{4} \cdot \text{dist}^2(AXX^T, \mu) + (1 + \frac{4}{\epsilon}) \cdot \left\| \tilde{A}_m X X^T - AXX^T \right\|_F^2$$

By A.21, $\text{dist}^2(AXX^T, \mu) \leq \text{dist}^2(A, \mu)$. Finally, we combining the last two inequalities with (A.24):

$$\left| \left( \text{dist}^2(\tilde{A}_m, \mu) + \left\| A - \tilde{A}_m \right\|_F^2 \right) - \text{dist}^2(A, \mu) \right|$$

$$\leq \frac{\varepsilon^2}{8} \cdot \text{dist}^2(A, \mu) + \frac{\epsilon}{4} \cdot \text{dist}^2(A, \mu) + \frac{\epsilon^2}{8} \cdot (1 + \frac{4}{\epsilon}) \cdot \text{dist}^2(A, \mu)$$

$$\leq \epsilon \cdot \text{dist}^2(A, \mu),$$

where in the last inequality we used the assumption $\epsilon \leq 1$. $\qquad \square$

**Corollary A.14.** *Assume we have $A \in \mathbb{R}^{n \times d}$ and CountSketch $S \in \mathbb{R}^{O(\mathrm{poly}(k/\epsilon)) \times n}$. Then, define $U \in \mathbb{R}^{d \times O(\mathrm{poly}(k/\epsilon))}$ with orthogonal columns spanning $row(SA)$. Also define $A^U = \pi_U(A)$ and $\mu_U$ as the set of optimal cluster centers found on $A^U$. Now, assume $\epsilon \in (0, 1/3]$. Then, $\mu_U$ is a $(1 + \epsilon)$-approximation to the optimal $k$-means clustering of $A$. That is, defining $\mu_U^*$ as the cluster centers which minimize the cost of partition $C_U$ on A, we have:*

$$\mathrm{dist}^2(A, \mu_U) \leq (1 + \epsilon) \, \mathrm{dist}^2(A, \mu_U^*)$$

*Proof:* By using $\frac{\epsilon}{3}$ in Theorem A.13 with $j$ as $k$,

$$\left| \mathrm{dist}^2(A^U, \mu_U) + \left\| A - A^U \right\|_F^2 - \mathrm{dist}^2(A, \mu_U) \right| \leq \frac{\epsilon}{3} \, \mathrm{dist}^2(A, \mu_U)$$

which implies that

$$(1 - \frac{\epsilon}{3}) \, \mathrm{dist}^2(A, \mu_U) \leq \mathrm{dist}^2(A^U, \mu_U) + \left\| A - A^U \right\|_F^2 \tag{A.25}$$

Likewise, by Theorem A.13 on $A^U$ and $\mu_U^*$ (and taking $j$ as $k$),

$$\left| \mathrm{dist}^2(A^U, \mu_U^*) + \left\| A - A^U \right\|_F^2 - \mathrm{dist}^2(A, \mu_U^*) \right| \leq \frac{\epsilon}{3} \, \mathrm{dist}^2(A, \mu_U^*)$$

which implies that

$$\mathrm{dist}^2(A^U, \mu_U^*) + \left\| A - A^U \right\|_F^2 \leq (1 + \frac{\epsilon}{3}) \, \mathrm{dist}^2(A, \mu_U^*) \tag{A.26}$$

By (A.25) and (A.26) together, we have:

$$(1 - \frac{\epsilon}{3}) \, \mathrm{dist}^2(A, \mu_U) \leq \mathrm{dist}^2(A^U, \mu_U) + \left\| A - A^U \right\|_F^2$$
$$\leq \mathrm{dist}^2(A^U, \mu_U^*) + \left\| A - A^U \right\|_F^2$$
$$\leq (1 + \frac{\epsilon}{3}) \, \mathrm{dist}^2(A, \mu_U^*)$$

Now, $\frac{1 + \epsilon/3}{1 - \epsilon/3} \leq 1 + \epsilon$, so we have $\mathrm{dist}^2(A, \mu_U) \leq (1 + \epsilon) \, \mathrm{dist}^2(A, \mu_U^*)$. $\qquad \square$

# B  SPECTRAL NORM GUARANTEE FOR ZIPFIAN MATRICES

In this section, we show that if the singular values of the input matrix $A$ follow a Zipfian distribution (i.e., $\sigma_i \propto i^{-\alpha}$ for a constant $\alpha$), we can find a $(1 + \epsilon)$ rank-$k$ approximation of $A$ with respect to the *spectral* norm. A key theorem in this section is the following.

**Theorem B.1 (Cohen et al. (2015), Theorem 27).** *Given an input matrix $A \in \mathbb{R}^{n \times d}$, there exists an algorithm that runs in $O(\mathrm{nnz}(A) + (n + d) \, \mathrm{poly}(k/\epsilon))$ and returns a projection matrix $P = QQ^\top$ such that with constant probability the following holds:*

$$\left\| AP - A \right\|_2^2 \leq (1 + \epsilon) \left\| A - A_k \right\|_2^2 + O(\frac{\epsilon}{k}) \left\| A - A_k \right\|_F^2 \tag{B.1}$$

Next, we prove the main claim in this section. There are various ways to prove this, using, for example, a bound on the *s*table rank of $A$; we give the following short proof for completeness.

**Theorem B.2.** *Given a matrix $A \in \mathbb{R}^{n \times d}$ whose singular values follow a Zipfian distribution (i.e., $\sigma_i \propto i^{-\alpha}$) with a constant $\alpha \geq 1/2$, there exists an algorithm that computes a rank-$k$ matrix $B$ (in factored form) such that $\left\| A - B \right\|_2^2 \leq (1 + \epsilon) \left\| A - A_k \right\|_2^2$.*

*Proof:* Note that since the singular values of $A$ follow a Zipfian distribution with parameter $\alpha$, for any value of $k$,

$$\left\| A - A_k \right\|_F^2 = \sum_{i=k+1}^{\mathrm{rank}(A)} \sigma_i^2 = C \cdot \sum_{i=k+1}^{\mathrm{rank}(A)} i^{-2\alpha} \qquad \qquad \triangleright \sigma_i = \sqrt{C}/i^\alpha$$
$$\leq C \cdot \int_k^{\mathrm{rank}(A)} x^{-2\alpha} \, dx$$

$$
\begin{aligned}
&\le k \cdot \frac{1}{2\alpha - 1} \cdot (1 + \frac{1}{k})^{2\alpha} \cdot C/(k+1)^{2\alpha} \\
&= O(k \cdot \sigma_{k+1}^2) \\
&= O(k \cdot \|A - A_k\|_2^2) \quad\quad\quad\quad\quad\quad\quad\quad\quad\quad\quad (\text{B.2})
\end{aligned}
$$

By an application of Theorem B.1, we can compute a matrix $B$ in a factored form in time $O(\mathrm{nnz}(A) + (n + d)\,\mathrm{poly}(k/\epsilon))$ such that with constant probability,

$$
\|B - A\|_2^2 \le (1 + \epsilon)\,\|A - A_k\|_2^2 + O(\tfrac{\epsilon}{k})\,\|A - A_k\|_F^2 \quad\quad \triangleright \text{ By Eq. (B.1)}
$$

$$
\le (1 + O(\epsilon))\,\|A - A_k\|_2^2 \quad\quad\quad\quad\quad\quad\quad \triangleright \text{ Eq. (B.2)} \quad\quad \square
$$

## C  GREEDY INITIALIZATION

In this section, we analyze the performance of the greedy algorithm on the two distributions mentioned in Theorem 4.5.

**Preliminaries and Notation.**  Left-multiplying $A$ by CountSketch $S \in \mathbb{R}^{m \times n}$ is equivalent to hashing the rows of $A$ to $m$ bins with coefficients in $\{-1, 1\}$. The greedy algorithm proceeds through the rows of $A$ (in some order) and decides which bin to hash to, denoting this by adding an entry to $S$. We will denote the bins as $b_i$ and their summed contents as $w_i$.

### C.1  SPIKED COVARIANCE MODEL WITH SPARSE LEFT SINGULAR VECTORS.

To recap, every matrix $A \in \mathbb{R}^{n \times d}$ from the distribution $\mathcal{A}_{sp}(s, \ell)$ has $s < k$ "heavy" rows $(A_{r_1}, \cdots, A_{r_s})$ of norm $\ell > 1$. The indices of the heavy rows can be arbitrary, but must be the same for all members of the distribution and are unknown to the algorithm. The remaining rows (called "light" rows) have unit norm.

In other words: let $\mathcal{R} = \{r_1, \ldots, r_s\}$. For all rows $A_i, i \in [n]$:

$$
A_i = \begin{cases} \ell \cdot v_i & \text{if } i \in \mathcal{R} \\ v_i & \text{o.w.} \end{cases}
$$

where $v_i$ is a uniformly random unit vector.

We also assume that $S_r, S_g \in \mathbb{R}^{k \times n}$ and non-increasing row norm ordering for the greedy algorithm.

**Proof sketch.**  First, we show that the greedy algorithm using a non-increasing row norm ordering will isolate heavy rows (i.e., each is alone in a bin). Then, we conclude by showing that this yields a better $k$-rank approximation error when $d$ is sufficiently large compared to $n$. We begin with some preliminary observations that will be of use later.

It is well known that a set of uniformly random vectors are $\epsilon$-*almost orthogonal* (i.e., the magnitudes of their pairwise inner products are at most $\epsilon$).

**Observation C.1.** *Let $v_1, \cdots, v_n \in \mathbb{R}^d$ be a set of random unit vectors. Then with high probability* $|\langle v_i, v_j \rangle| \le 2\sqrt{\frac{\log n}{d}}, \forall\, i < j \le n.$

We define $\bar{\epsilon} = 2\sqrt{\frac{\log n}{d}}$.

**Observation C.2.** *Let $u_1, \cdots, u_t$ be a set of vectors such that for each pair of $i < j \le t$, $|\langle \frac{u_i}{\|u_i\|}, \frac{u_j}{\|u_j\|} \rangle| \le \epsilon$, and $g_i, \cdots, g_j \in \{-1, 1\}$. Then,*

$$
\sum_{i=1}^t \|u_i\|_2^2 - 2\epsilon \sum_{i<j\le t} \|u_i\|_2 \|u_j\|_2 \le \left\| \sum_{i=1}^t g_i u_i \right\|_2^2 \le \sum_{i=1}^t \|u_i\|_2^2 + 2\epsilon \sum_{i<j\le t} \|u_i\|_2 \|u_j\|_2 \quad (\text{C.1})
$$

Next, a straightforward consequence of $\epsilon$-almost orthogonality is that we can find a QR-factorization of the matrix of such vectors where $R$ (an upper diagonal matrix) has diagonal entries close to 1 and entries above the diagonal are close to 0.

**Lemma C.3.** *Let $u_1, \cdots, u_t \in \mathbb{R}^d$ be a set of unit vectors such that for any pair of $i < j \leq t$, $|\langle u_i, u_j \rangle| \leq \epsilon$ where $\epsilon = O(t^{-2})$. There exists an orthonormal basis $e_1, \cdots, e_t$ for the subspace spanned by $u_1, \cdots, u_t$ such that for each $i \leq t$, $u_i = \sum_{j=1}^{i} a_{i,j} e_j$ where $a_{i,i}^2 \geq 1 - \sum_{j=1}^{i-1} j^2 \cdot \epsilon^2$ and for each $j < i$, $a_{i,j}^2 \leq j^2 \epsilon^2$.*

*Proof:* We follow the Gram-Schmidt process to construct the orthonormal basis $e_1, \cdots, e_t$ of the space spanned by $u_1, \cdots, u_t$. by first setting $e_1 = u_1$ and then processing $u_2, \cdots, u_t$, one by one.

The proof is by induction. We show that once the first $j$ vectors $u_1, \cdots, u_j$ are processed the statement of the lemma holds for these vectors. Note that the base case of the induction trivially holds as $u_1 = e_1$. Next, suppose that the induction hypothesis holds for the first $\ell$ vectors $u_1, \cdots, u_\ell$.

**Claim C.4.** *For each $j \leq \ell$, $a_{\ell+1,j}^2 \leq j^2 \epsilon^2$.*

*Proof:* The proof of the claim is itself by induction. Note that, for $j = 1$ and using the fact that $|\langle u_1, u_{\ell+1} \rangle| \leq \epsilon$, the statement holds and $a_{\ell+1,1}^2 \leq \epsilon^2$. Next, suppose that the statement holds for all $j \leq i < \ell$, then by $|\langle u_{i+1}, u_{\ell+1} \rangle| \leq \epsilon$,

$$|a_{\ell+1,i+1}| \leq (|\langle u_{\ell+1}, u_{i+1} \rangle| + \sum_{j=1}^{i} |a_{\ell+1,j}| \cdot |a_{i+1,j}|)/|a_{i+1,i+1}|$$

$$\leq (\epsilon + \sum_{j=1}^{i} j^2 \epsilon^2)/|a_{i+1,i+1}| \quad \triangleright \text{ by induction hypothesis on } a_{\ell+1,j} \text{ for } j \leq i$$

$$\leq (\epsilon + \sum_{j=1}^{i} j^2 \epsilon^2)/(1 - \sum_{j=1}^{i} j^2 \cdot \epsilon^2)^{1/2} \quad \triangleright \text{ by induction hypothesis on } a_{i+1,i+1}$$

$$\leq (\epsilon + \sum_{j=1}^{i} j^2 \epsilon^2) \cdot (1 - \sum_{j=1}^{i} j^2 \cdot \epsilon^2)^{1/2} \cdot (1 + 2 \cdot \sum_{j=1}^{i} j^2 \epsilon^2)$$

$$\leq (\epsilon + \sum_{j=1}^{i} j^2 \epsilon^2) \cdot (1 + 2 \cdot \sum_{j=1}^{i} j^2 \epsilon^2)$$

$$\leq \epsilon((\sum_{j=1}^{i} j^2 \epsilon) \cdot (1 + 4\epsilon \cdot \sum_{j=1}^{i} j^2 \epsilon) + 1)$$

$$\leq \epsilon(i+1) \quad \triangleright \text{ by } \epsilon = O(t^{-2}) \qquad \qquad \square$$

Finally, since $\|u_{\ell+1}\|_2^2 = 1$, $a_{\ell+1,\ell+1}^2 \geq 1 - \sum_{j=1}^{\ell} j^2 \epsilon^2$. $\qquad \square$

**Corollary C.5.** *Suppose that $\bar{\epsilon} = O(t^{-2})$. There exists an orthonormal basis $e_1, \cdots, e_t$ for the space spanned by the randomly picked vectors $v_1, \cdots, v_t$, of unit norm, so that for each $i$, $v_i = \sum_{j=1}^{i} a_{i,j} e_j$ where $a_{i,i}^2 \geq 1 - \sum_{j=1}^{i-1} j^2 \cdot \bar{\epsilon}^2$ and for each $j < i$, $a_{i,j}^2 \leq j^2 \cdot \bar{\epsilon}^2$.*

*Proof:* The proof follows from Lemma C.3 and the fact that the set of vectors $v_1, \cdots, v_t$ are $\bar{\epsilon}$-almost orthogonal (by Observation C.1). $\qquad \square$

The first main step is to show that the greedy algorithm (with non-increasing row norm ordering) will isolate rows into their own bins until all bins are filled. In particular, this means that the heavy rows (the first to be processed) will all be isolated.

We note that because we set $\mathrm{rank}(SA) = k$, the $k$-rank approximation cost is the simplified expression $\|AVV^\top - A\|_F^2$, where $U\Sigma V^\top = SA$, rather than $\|[AV]_k V^\top - A\|_F^2$. This is just the projection cost onto $row(SA)$. Also, we observe that minimizing this projection cost is the same as maximizing the sum of squared projection coefficients:

$$\min_S \|A - AVV^\top\|_F^2 \sim \min_S \sum_{i \in [n]} \|A_i - (\langle A_i, v_1 \rangle v_1 + \ldots + \langle A_i, v_k \rangle v_k)\|_2^2$$

$$\sim \min_S \sum_{i \in [n]} (\|A_i\|_2^2 - \sum_{j \in [k]} \langle A_i, v_j \rangle^2)$$

$$\sim \max_S \sum_{i \in [n]} \sum_{j \in [k]} \langle A_i, v_j \rangle^2$$

In the following sections, we will prove that our greedy algorithm makes certain choices by showing that these choices maximize the sum of squared projection coefficients.

**Lemma C.6.** *For any matrix $A$ or batch of matrices $\mathcal{A}$, at the end of iteration $k$, the learned CountSketch matrix $S$ maps each row to an isolated bin. In particular, heavy rows are mapped to isolated bins.*

*Proof:* For any iteration $i \leq k$, we consider the choice of assigning $A_i$ to an empty bin versus an occupied bin. Without loss of generality, let this occupied bin be $b_{i-1}$, which already contains $A_{i-1}$.

We consider the difference in cost for empty versus occupied. We will do this cost comparison for $A_j$ with $j \leq i - 2$, $j \geq i + 1$, and finally, $j \in \{i - 1, i\}$.

First, we let $\{e_1, \dots, e_i\}$ be an orthonormal basis for $\{A_1, \dots, A_i\}$ such that for each $r \leq i$, $A_r = \sum_{j=1}^r a_{r,j} e_j$ where $a_{r,r} > 0$. This exists by Lemma C.3. Let $\{e_1, \dots, e_{i-2}, \overline{e}\}$ be an orthonormal basis for $\{A_1, \dots, A_{i+2}, A_{i-1} \pm A_i\}$. Now, $\overline{e} = c_0 e_{i-1} + c_1 e_i$ for some $c_0, c_1$ because $(A_{i-1} \pm A_i) - \text{proj}_{\{e_1, \dots, e_{i-2}\}}(A_{i-1} \pm A_i) \in \text{span}(e_{i-1}, e_i)$. We note that $c_0^2 + c_1^2 = 1$ because we let $\overline{e}$ be a unit vector. We can find $c_0, c_1$ to be:

$$c_0 = \frac{a_{i-1,i-1} + a_{i,i-1}}{\sqrt{(a_{i-1,i-1} + a_{i,i-1})^2 + a_{i,i}^2}}, \quad c_1 = \frac{a_{i,i}}{\sqrt{(a_{i-1,i-1} + a_{i,i-1})^2 + a_{i,i}^2}}$$

1. $j \leq i - 2$: The cost is zero for both cases because $A_j \in \text{span}(\{e_1, \dots, e_{i-2}\})$.

2. $j \geq i + 1$: We compare the rewards (sum of squared projection coefficients) and find that $\{e_1, \dots, e_{i-2}, \overline{e}\}$ is no better than $\{e_1, \dots, e_i\}$.

$$\begin{aligned}
\langle A_j, \overline{e} \rangle^2 &= (c_0 \langle A_j, e_{i-1} \rangle + c_1 \langle A_j, e_i \rangle)^2 \\
&\leq (c_1^2 + c_0^2)(\langle A_j, e_{i-1} \rangle^2 + \langle A_j, e_i \rangle^2) \qquad \triangleright \text{ Cauchy-Schwarz inequality} \\
&= \langle A_j, e_{i-1} \rangle^2 + \langle A_j, e_i \rangle^2
\end{aligned}$$

3. $j \in \{i - 1, i\}$: We compute the sum of squared projection coefficients of $A_{i-1}$ and $A_i$ onto $\overline{e}$:

$$\begin{aligned}
&(\frac{1}{(a_{i-1,i-1} + a_{i,i-1})^2 + a_{i,i}^2}) \cdot (a_{i-1,i-1}^2 (a_{i-1,i-1} + a_{i,i-1})^2 \\
&\quad + (a_{i,i-1}(a_{i-1,i-1} + a_{i,i-1}) + a_{i,i} a_{i,i})^2) \\
=&(\frac{1}{(a_{i-1,i-1} + a_{i,i-1})^2 + a_{i,i}^2}) \cdot ((a_{i-1,i-1} + a_{i,i-1})^2 (a_{i-1,i-1}^2 \qquad \text{(C.2)} \\
&\quad + a_{i,i-1}^2) + a_{i,i}^4 + 2 a_{i,i-1} a_{i,i}^2 (a_{i-1,i-1} + a_{i,i-1})) \qquad \text{(C.3)}
\end{aligned}$$

On the other hand, the sum of squared projection coefficients of $A_{i-1}$ and $A_i$ onto $e_{i-1} \cup e_i$ is:

$$(\frac{(a_{i-1,i-1} + a_{i,i-1})^2 + a_{i,i}^2}{(a_{i-1,i-1} + a_{i,i-1})^2 + a_{i,i}^2}) \cdot (a_{i-1,i-1}^2 + a_{i,i-1}^2 + a_{i,i}^2) \qquad \text{(C.4)}$$

Hence, the difference between the sum of squared projections of $A_{i-1}$ and $A_i$ onto $\overline{e}$ and $e_{i-1} \cup e_i$ is ((C.4) - (C.3))

$$\begin{aligned}
&\frac{a_{i,i}^2((a_{i-1,i-1} + a_{i,i-1})^2 + a_{i-1,i-1}^2 + a_{i,i-1}^2 - 2 a_{i,i-1}(a_{i-1,i-1} + a_{i,i-1}))}{((a_{i-1,i-1} + a_{i,i-1})^2 + a_{i,i}^2)} \\
=& \frac{2 a_{i,i}^2 a_{i-1,i-1}^2}{((a_{i-1,i-1} + a_{i,i-1})^2 + a_{i,i}^2)} > 0
\end{aligned}$$

Thus, we find that $\{e_1, \dots, e_i\}$ is a strictly better basis than $\{e_1, \dots, e_{i-2}, \overline{e}\}$. This means the greedy algorithm will choose to place $A_i$ in an empty bin. $\qquad \square$

Next, we show that none of the rows left to be processed (all light rows) will be assigned to the same bin as a heavy row. The main proof idea is to compare the cost of "colliding" with a heavy row to the cost of "avoiding" the heavy rows. Specifically, we compare the *decrease* (before and after bin assignment of a light row) in the sum of squared projection coefficients, lower-bounding it in the former case and upper-bounding it in the latter.

We introduce some results that will be used in Lemma C.10.

**Claim C.7.** *Let $A_{k+r}, r \in [1, \ldots, n-k]$ be a light row not yet processed by the greedy algorithm. Let $\{e_1, \ldots, e_k\}$ be the Gram-Schmidt basis for the current $\{w_1, \ldots, w_k\}$. Let $\beta = O(n^{-1}k^{-3})$ upper bound the inner products of normalized $A_{k+r}, w_1, \ldots, w_k$. Then, for any bin $i$, $\langle e_i, A_{k+r} \rangle^2 \leq \beta^2 \cdot k^2$.*

*Proof:* This is a straightforward application of Lemma C.3. From that, we have $\langle A_{k+r}, e_i \rangle^2 \leq i^2 \beta^2$, for $i \in [1, \ldots, k]$, which means $\langle A_{k+r}, e_i \rangle^2 \leq k^2 \beta^2$. $\qquad\square$

**Claim C.8.** *Let $A_{k+r}$ be a light row that has been processed by the greedy algorithm. Let $\{e_1, \ldots, e_k\}$ be the Gram-Schmidt basis for the current $\{w_1, \ldots, w_k\}$. If $A_{k+r}$ is assigned to bin $b_{k-1}$ (w.l.o.g.), the squared projection coefficient of $A_{k+r}$ onto $e_i, i \neq k-1$ is at most $4\beta^2 \cdot k^2$, where $\beta = O(n^{-1}k^{-3})$ upper bounds the inner products of normalized $A_{k+r}, w_1, \cdots, w_k$.*

*Proof:* Without loss of generality, it suffices to bound the squared projection of $A_{k+r}$ onto the direction of $w_k$ that is orthogonal to the subspace spanned by $w_1, \cdots, w_{k-1}$. Let $e_1, \cdots, e_k$ be an orthonormal basis of $w_1, \cdots, w_k$ guaranteed by Lemma C.3. Next, we expand the orthonormal basis to include $e_{k+1}$ so that we can write the normalized vector of $A_{k+r}$ as $v_{k+r} = \sum_{j=1}^{k+1} b_j e_j$. By a similar approach to the proof of Lemma C.3, for each $j \leq k-2$, $b_j \leq \beta^2 j^2$. Next, since $|\langle w_k, v_{k+r} \rangle| \leq \beta$,

$$|b_k| \leq \frac{1}{|\langle w_k, e_k \rangle|} \cdot \left( |\langle w_k, v_{k+r} \rangle| + \sum_{j=1}^{k-1} |b_j \cdot \langle w_k, e_j \rangle| \right)$$

$$\leq \frac{1}{\sqrt{1 - \sum_{j=1}^{k-1} \beta^2 \cdot j^2}} \cdot \left( \beta + \sum_{j=1}^{k-2} \beta^2 \cdot j^2 + (k-1) \cdot \beta \right) \quad \triangleright |b_{k-1}| \leq 1$$

$$= \frac{\beta + \sum_{j=1}^{k-2} \beta^2 \cdot j^2}{\sqrt{1 - \sum_{j=1}^{k-1} \beta^2 \cdot j^2}} + (k-1)\beta$$

$$\leq 2(k-1)\beta - \frac{\beta^2 (k-1)^2}{\sqrt{1 - \sum_{j=1}^{k-1} \beta^2 \cdot j^2}} \quad\quad\quad \triangleright \text{similar to the proof of Lemma C.3}$$

$$< 2\beta \cdot k \qquad\qquad\qquad\qquad\qquad\qquad\qquad\qquad\qquad\qquad\qquad \blacksquare$$

Hence, the squared projection of $A_{k+r}$ onto $e_k$ is at most $4\beta^2 \cdot k^2 \cdot \|A_{k+r}\|_2^2$. We assumed $\|A_{k+r}\|$. $\square$

**Claim C.9.** *We assume that the absolute values of the inner products of vectors in $v_1, \cdots, v_n$ are at most $\bar{\epsilon} < 1/(n^2 \sum_{A_i \in b} \|A_i\|_2)$ and the absolute values of the inner products of the normalized vectors of $w_1, \cdots, w_k$ are at most $\beta = O(n^{-3}k^{-\frac{3}{2}})$. Suppose that bin $b$ contains the row $A_{k+r}$. Then, the squared projection of $A_{k+r}$ onto the direction of $w$ orthogonal to $\text{span}(\{w_1, \cdots, w_k\} \setminus \{w\})$ is at most $\frac{\|A_{k+r}\|_2^4}{\|w\|_2^2} + O(n^{-2})$ and is at least $\frac{\|A_{k+r}\|_2^4}{\|w\|_2^2} - O(n^{-2})$.*

*Proof:* Without loss of generality, we assume that $A_{k+r}$ is mapped to $b_k$; $w = w_k$. First, we provide an upper and a lower bound for $|\langle v_{k+r}, \overline{w}_k \rangle|$ where for each $i \leq k$, we let $\overline{w}_i = \frac{w_i}{\|w_i\|_2}$ denote the normalized vector of $w_i$. Recall that by definition $v_{k+r} = \frac{A_{k+r}}{\|A_{k+r}\|_2}$.

$$|\langle \overline{w}_k, v_{k+r} \rangle| \leq \frac{\|A_{k+r}\|_2 + \sum_{A_i \in b_k} \bar{\epsilon} \|A_i\|_2}{\|w_k\|_2}$$

$$\leq \frac{\|A_{k+r}\|_2 + n^{-2}}{\|w_k\|_2} \quad\quad\quad \triangleright \text{ by } \bar{\epsilon} < \frac{n^{-2}}{\sum_{A_i \in b_k} \|A_i\|_2}$$

$$\leq \frac{\|A_{k+r}\|_2}{\|w_k\|_2} + n^{-2} \quad\quad\quad \triangleright \|w_k\|_2 \geq 1 \qquad\qquad (C.5)$$

$$|\langle \overline{w}_k, v_{k+r}\rangle| \geq \frac{\|A_{k+r}\|_2 - \sum_{A_i \in b_k} \|A_i\|_2 \cdot \overline{\epsilon}}{\|w_k\|_2}$$

$$\geq \frac{\|A_{k+r}\|_2}{\|w_k\|_2} - n^{-2} \tag{C.6}$$

Now, let $\{e_1, \cdots, e_k\}$ be an orthonormal basis for the subspace spanned by $\{w_1, \cdots, w_k\}$ guaranteed by Lemma C.3. Next, we expand the orthonormal basis to include $e_{k+1}$ so that we can write $v_{k+r} = \sum_{j=1}^{k+1} b_j e_j$. By a similar approach to the proof of Lemma C.3, we can show that for each $j \leq k-1$, $b_j^2 \leq \beta^2 j^2$. Moreover,

$$|b_k| \leq \frac{1}{|\langle \overline{w}_k, e_k\rangle|} \cdot \left( |\langle \overline{w}_k, v_{k+r}\rangle| + \sum_{j=1}^{k-1} |b_j \cdot \langle \overline{w}_k, e_j\rangle| \right)$$

$$\leq \frac{1}{\sqrt{1 - \sum_{j=1}^{k-1} \beta^2 \cdot j^2}} \cdot \left( |\langle \overline{w}_k, v_{k+r}\rangle| + \sum_{j=1}^{k-1} \beta^2 \cdot j^2 \right) \qquad \triangleright \text{ by Lemma C.3}$$

$$\leq \frac{1}{\sqrt{1 - \sum_{j=1}^{k-1} \beta^2 \cdot j^2}} \cdot \left( n^{-2} + \frac{\|A_{k+r}\|_2}{\|w_k\|_2} + \sum_{j=1}^{k-1} \beta^2 \cdot j^2 \right) \qquad \triangleright \text{ by (C.5)}$$

$$< \beta \cdot k + \frac{1}{\sqrt{1 - \beta^2 k^3}} \cdot \left( n^{-2} + \frac{\|A_{k+r}\|_2}{\|w_k\|_2} \right) \qquad \triangleright \text{ similar to the proof of Lemma C.3}$$

$$\leq O(n^{-2}) + (1 + O(n^{-2})) \frac{\|A_{k+r}\|_2}{\|w_k\|_2} \qquad \triangleright \text{ by } \beta = O(n^{-3} k^{-\frac{3}{2}})$$

$$\leq \frac{\|A_{k+r}\|_2}{\|w_k\|_2} + O(n^{-2}) \qquad \triangleright \frac{\|A_{k+r}\|_2}{\|w_k\|_2} \leq 1 \qquad \blacksquare$$

and,

$$|b_k| \geq \frac{1}{|\langle \overline{w}_k, e_k\rangle|} \cdot \left( |\langle \overline{w}_k, v_{k+r}\rangle| - \sum_{j=1}^{k-1} |b_j \cdot \langle \overline{w}_k, e_j\rangle| \right)$$

$$\geq |\langle \overline{w}_k, v_{k+r}\rangle| - \sum_{j=1}^{k-1} \beta^2 \cdot j^2 \qquad \triangleright \text{ since } |\langle \overline{w}_k, e_k\rangle| \leq 1$$

$$\geq \frac{\|A_{k+r}\|_2}{\|w_k\|_2} - n^{-2} - \sum_{j=1}^{k-1} \beta^2 \cdot j^2 \qquad \triangleright \text{ by (C.6)}$$

$$\geq \frac{\|A_{k+r}\|_2}{\|w_k\|_2} - O(n^{-2}) \qquad \triangleright \text{ by } \beta = O(n^{-3} k^{-\frac{3}{2}})$$

Hence, the squared projection of $A_{k+r}$ onto $e_k$ is at most $\frac{\|A_{k+r}\|_2^4}{\|w_k\|_2^2} + O(n^{-2})$ and is at least $\frac{\|A_{k+r}\|_2^4}{\|w_k\|_2^2} - O(n^{-2})$. $\qquad \square$

Now, we show that at the end of the algorithm no light row will be assigned to the bins that contain heavy rows.

**Lemma C.10.** *We assume that the absolute values of the inner products of vectors in $v_1, \cdots, v_n$ are at most $\overline{\epsilon} < \min\{n^{-2} k^{-\frac{5}{3}}, (n \sum_{A_i \in w} \|A_i\|_2)^{-1}\}$. At each iteration $k + r$, the greedy algorithm will assign the light row $A_{k+r}$ to a bin that does not contain a heavy row.*

*Proof:* The proof is by induction. Lemma C.6 implies that no light row has been mapped to a bin that contains a heavy row for the first $k$ iterations. Next, we assume that this holds for the first $k + r - 1$ iterations and show that is also must hold for the $(k + r)$-th iteration.

To this end, we compare the sum of squared projection coefficients when $A_{k+r}$ avoids and collides with a heavy row.

First, we upper bound $\beta = \max_{i \neq j \leq k} |\langle w_i, w_j \rangle| / (\|w_i\|_2 \|w_j\|_2)$. Let $c_i$ and $c_j$ respectively denote the number of rows assigned to $b_i$ and $b_j$.

$$\beta = \max_{i \neq j \leq k} \frac{|\langle w_i, w_j \rangle|}{\|w_i\|_2 \|w_j\|_2} \leq \frac{c_i \cdot c_j \cdot \overline{\epsilon}}{\sqrt{c_i - 2\overline{\epsilon}c_i^2} \cdot \sqrt{c_j - 2\overline{\epsilon}c_j^2}} \qquad \triangleright \text{ Observation C.2}$$

$$\leq 16\overline{\epsilon}\sqrt{c_i c_j} \qquad \triangleright \overline{\epsilon} \leq n^{-2}k^{-5/3}$$

$$\leq n^{-1}k^{-\frac{5}{3}} \qquad \triangleright \overline{\epsilon} \leq n^{-2}k^{-5/3}$$

**1. If $A_{k+r}$ is assigned to a bin that contains $c$ light rows and no heavy rows.** In this case, the projection loss of the heavy rows $A_1, \cdots, A_s$ onto $row(SA)$ remains zero. Thus, we only need to bound the change in the sum of squared projection coefficients of the light rows before and after iteration $k + r$.

Without loss of generality, let $w_k$ denote the bin that contains $A_{k+r}$. Since $\mathbb{S}_{k-1} = \text{span}(\{w_1, \cdots, w_{k-1}\})$ has not changed, we only need to bound the difference in cost between projecting onto the component of $w_k - A_{k+r}$ orthogonal to $\mathbb{S}_{k-1}$ and the component of $w_k$ orthogonal to $\mathbb{S}_{k-1}$, respectively denoted as $e_k$ and $\overline{e}_k$.

I. By Claim C.7, for the light rows that are not yet processed (i.e., $A_j$ for $j > k + r$), the squared projection of each onto $e_k$ is at most $\beta^2 k^2$. Hence, the total decrease in the squared projection is at most $(n - k - r) \cdot \beta^2 k^2$.

II. By Claim C.8, for the processed light rows that are not mapped to the last bin, the squared projection of each onto $e_k$ is at most $4\beta^2 k^2$. Hence, the total decrease in the squared projection cost is at most $(r - 1) \cdot 4\beta^2 k^2$.

III. For each row $A_i \neq A_{k+r}$ that is mapped to the last bin, by Claim C.9 and the fact $\|A_i\|_2^4 = \|A_i\|_2^2 = 1$, the squared projection of $A_i$ onto $e_k$ is at most $\frac{\|A_i\|_2^2}{\|w_k - A_{k+r}\|_2^2} + O(n^{-2})$ and the squared projection of $A_i$ onto $\overline{e}_k$ is at least $\frac{\|A_i\|_2^2}{\|w_k\|_2^2} - O(n^{-2})$.

Moreover, the squared projection of $A_{k+r}$ onto $e_k$ compared to $\overline{e}_k$ increases by at least $(\frac{\|A_{k+r}\|_2^2}{\|w_k\|_2^2} - O(n^{-2})) - O(n^{-2}) = \frac{\|A_{k+r}\|_2^2}{\|w_k\|_2^2} - O(n^{-2})$.

Hence, the total squared projection of the rows in the bin $b_k$ decreases by at least:

$$\left( \sum_{A_i \in w_k / \{A_{r+k}\}} \frac{\|A_i\|_2^2}{\|w_k - A_{r+k}\|_2^2} + O(n^{-2}) \right) - \left( \sum_{A_i \in w_k} \frac{\|A_i\|_2^2}{\|w_k\|_2^2} - O(n^{-2}) \right)$$

$$\leq \frac{\|w_k - A_{r+k}\|_2^2 + O(n^{-1})}{\|w_k - A_{r+k}\|_2^2} - \frac{\|w_k\|_2^2 - O(n^{-1})}{\|w_k\|_2^2} + O(n^{-1}) \qquad \triangleright \text{ by Observation C.2}$$

$$\leq O(n^{-1})$$

Hence, summing up the bounds in items I to III above, the total decrease in the sum of squared projection coefficients is at most $O(n^{-1})$.

**2. If $A_{k+r}$ is assigned to a bin that contains a heavy row.** Without loss of generality, we can assume that $A_{k+r}$ is mapped to $b_k$ that contains the heavy row $A_s$. In this case, the distance of heavy rows $A_1, \cdots, A_{s-1}$ onto the space spanned by the rows of $SA$ is zero. Next, we bound the amount of change in the squared distance of $A_s$ and light rows onto the space spanned by the rows of $SA$.

Note that the $(k - 1)$-dimensional space corresponding to $w_1, \cdots, w_{k-1}$ has not changed. Hence, we only need to bound the decrease in the projection distance of $A_{k+r}$ onto $\overline{e}_k$ compared to $e_k$ (where $\overline{e}_k, e_k$ are defined similarly as in the last part).

1. For the light rows other than $A_{k+r}$, the squared projection of each onto $e_k$ is at most $\beta^2 k^2$. Hence, the total increase in the squared projection of light rows onto $e_k$ is at most $(n - k) \cdot \beta^2 k^2 = O(n^{-1})$.

2. By Claim C.9, the sum of squared projections of $A_s$ and $A_{k+r}$ onto $e_k$ decreases by at least

$$\|A_s\|_2^2 - (\frac{\|A_s\|_2^4 + \|A_{k+r}\|_2^4}{\|A_s + A_{r+k}\|_2^2} + O(n^{-1}))$$

$$\geq \|A_s\|_2^2 - (\frac{\|A_s\|_2^4 + \|A_{k+r}\|_2^4}{\|A_s\|_2^2 + \|A_{r+k}\|_2^2 - n^{-O(1)}} + O(n^{-1})) \qquad \triangleright \text{ by Observation C.2}$$

$$\geq \frac{\|A_{r+k}\|_2^2 (\|A_s\|_2^2 - \|A_{k+r}\|_2^2) - \|A_s\|_2^2 \cdot O(n^{-1})}{\|A_s\|_2^2 + \|A_{r+k}\|_2^2 - O(n^{-1})} - O(n^{-1})$$

$$\geq \frac{\|A_{r+k}\|_2^2 (\|A_s\|_2^2 - \|A_{k+r}\|_2^2) - \|A_s\|_2^2 \cdot O(n^{-1})}{\|A_s\|_2^2 + \|A_{r+k}\|_2^2} - O(n^{-1})$$

$$\geq \frac{\|A_{r+k}\|_2^2 (\|A_s\|_2^2 - \|A_{k+r}\|_2^2)}{\|A_s\|_2^2 + \|A_{r+k}\|_2^2} - O(n^{-1})$$

$$\geq \frac{\|A_{r+k}\|_2^2 (1 - (\|A_{k+r}\|_2^2 / \|A_s\|_2^2))}{1 + (\|A_{r+k}\|_2^2 / \|A_s\|_2^2)} - O(n^{-1})$$

$$\geq \|A_{r+k}\|_2^2 (1 - \frac{\|A_{k+r}\|_2}{\|A_s\|_2}) - O(n^{-1}) \qquad \triangleright \frac{1-\epsilon^2}{1+\epsilon^2} \geq 1 - \epsilon$$

Hence, in this case, the total decrease in the squared projection is at least

$$\|A_{r+k}\|_2^2 (1 - \frac{\|A_{k+r}\|_2}{\|A_s\|_2}) - O(n^{-1}) = 1 - \frac{\|A_{k+r}\|_2}{\|A_s\|_2}) - O(n^{-1}) \qquad \triangleright \|A_{r+k}\|_2 = 1$$

$$= 1 - (1/\sqrt{\ell}) - O(n^{-1}) \qquad \triangleright \|A_s\|_2 = \sqrt{\ell}$$

Thus, for a sufficiently large value of $\ell$, the greedy algorithm will assign $A_{k+r}$ to a bin that only contains light rows. This completes the inductive proof and in particular implies that at the end of the algorithm, heavy rows are assigned to isolated bins. $\qquad \square$

**Corollary C.11.** *The approximation loss of the best* rank-$k$ *approximate solution in the rowspace* $S_g A$ *for* $A \sim \mathcal{A}_{sp}(s, \ell)$ *where* $A \in \mathbb{R}^{n \times d}$ *for* $d = \Omega(n^4 k^4 \log n)$ *and* $S_g$ *is the CountSketch constructed by the greedy algorithm with non-increasing order is at most* $n - s$.

*Proof:* First, we need to show that absolute values of the inner products of vectors in $v_1, \cdots, v_n$ is at most $\bar{\epsilon} < \min\{n^{-2}k^{-2}, (n \sum_{A_i \in w} \|A_i\|_2)^{-1}\}$ so that we can apply Lemma C.10. To show this, note that by Observation C.1, $\bar{\epsilon} \leq 2\sqrt{\frac{\log n}{d}} \leq n^{-2}k^{-2}$ since $d = \Omega(n^4 k^4 \log n)$. The proof follows from Lemma C.6 and Lemma C.10. Since all heavy rows are mapped to isolated bins, the projection loss of the light rows is at most $n - s$. $\qquad \square$

Next, we bound the Frobenius norm error of the best rank-$k$-approximation solution constructed by the standard CountSketch with a randomly chosen sparsity pattern.

**Lemma C.12.** *Let* $s = \alpha k$ *where* $0.7 < \alpha < 1$. *The expected squared loss of the best* rank-$k$ *approximate solution in the rowspace* $S_r A$ *for* $A \in \mathbb{R}^{n \times d} \sim \mathcal{A}_{sp}(s, \ell)$ *where* $d = \Omega(n^6 \ell^2)$ *and* $S_r$ *is the sparsity pattern of CountSketch is chosen uniformly at random is at least* $n + \frac{\ell k}{4e} - (1 + \alpha)k - n^{-O(1)}$.

*Proof:* We can interpret the randomized construction of the CountSketch as a "balls and bins" experiment. In particular, considering the heavy rows, we compute the expected number of bins (i.e., rows in $S_r A$) that contain a heavy row. Note that the expected number of rows in $S_r A$ that do not contain any heavy row is $k \cdot (1 - \frac{1}{k})^s \geq k \cdot e^{-\frac{s}{k-1}}$. Hence, the number of rows in $S_r A$ that contain a heavy row of $A$ is at most $k(1 - e^{-\frac{s}{k-1}})$. Thus, at least $s - k(1 - e^{-\frac{s}{k-1}})$ heavy rows are not mapped to an isolated bin (i.e., they collide with some other heavy rows). Then, it is straightforward to show that the squared loss of each such row is at least $\ell - n^{-O(1)}$.

**Claim C.13.** *Suppose that heavy rows* $A_{r_1}, \cdots, A_{r_c}$ *are mapped to the same bin via a CountSketch* $S$. *Then, the total squared distances of these rows from the subspace spanned by* $SA$ *is at least* $(c - 1)\ell - O(n^{-1})$.

*Proof:* Let $b$ denote the bin that contains the rows $A_{r_1}, \cdots, A_{r_c}$ and suppose that it has $c'$ light rows as well. Note that by Claim C.8 and Claim C.9, the squared projection of each row $A_{r_i}$ onto the subspace spanned by the $k$ bins is at most

$$\frac{\|A_{h_i}\|_2^4}{\|w\|_2^2} + O(n^{-1})$$

$$\leq \frac{\ell^2}{c\ell + c' - 2\bar{\epsilon}(c^2\ell + cc'\sqrt{\ell} + c'^2)} + O(n^{-1})$$

$$\leq \frac{\ell^2}{c\ell - n^{-O(1)}} + n^{-O(1)} \qquad\qquad \triangleright \text{ by } \bar{\epsilon} \leq n^{-3}\ell^{-1}$$

$$\leq \frac{\ell^2}{c^2\ell^2} \cdot (c\ell + O(n^{-1}) + O(n^{-1})$$

$$\leq \frac{\ell}{c} + O(n^{-1})$$

Hence, the total squared loss of these $c$ heavy rows is at least $c\ell - c \cdot (\frac{\ell}{c} + O(n^{-1})) \geq (c-1)\ell - O(n^{-1})$. $\qquad\square$

Hence, the expected total squared loss of heavy rows is at least:

$$\ell \cdot (s - k(1 - e^{-\frac{s}{k-1}})) - s \cdot n^{-O(1)}$$

$$\geq \ell \cdot k(\alpha - 1 + e^{-\alpha}) - \ell\alpha - n^{-O(1)} \qquad\qquad \triangleright s = \alpha \cdot (k-1) \text{ where } 0.7 < \alpha < 1$$

$$\geq \frac{\ell k}{2e} - \ell - n^{-O(1)} \qquad\qquad\qquad\qquad \triangleright \alpha \geq 0.7$$

$$\geq \frac{\ell k}{4e} - O(n^{-1}) \qquad\qquad\qquad\qquad\quad \triangleright \text{ assuming } k > 4e$$

Next, we compute a lower bound on the expected squared loss of the light rows. Note that Claim C.8 and Claim C.9 imply that when a light row collides with other rows, its contribution to the total squared loss (which the loss accounts for the amount it decreases from the squared projection of the other rows in the bin as well) is at least $1 - O(n^{-1})$. Hence, the expected total squared loss of the light rows is at least:

$$(n - s - k)(1 - O(n^{-1})) \geq (n - (1 + \alpha) \cdot k) - O(n^{-1})$$

Hence, the expected squared loss of a CountSketch whose sparsity is picked at random is at least

$$\frac{\ell k}{4e} - O(n^{-1}) + n - (1 + \alpha)k - O(n^{-1}) \geq n + \frac{\ell k}{4e} - (1 + \alpha)k - O(n^{-1}) \qquad\quad \square$$

**Corollary C.14.** *Let $s = \alpha(k-1)$ where $0.7 < \alpha < 1$ and let $\ell \geq \frac{(4e+1)n}{\alpha k}$. Let $S_g$ be the CountSketch whose sparsity pattern is learned over a training set drawn from $\mathcal{A}_{sp}$ via the greedy approach. Let $S_r$ be a CountSketch whose sparsity pattern is picked uniformly at random. Then, for an $n \times d$ matrix $A \sim \mathcal{A}_{sp}$ where $d = \Omega(n^6\ell^2)$, the expected loss of the best rank-$k$ approximation of $A$ returned by $S_r$ is worse than the approximation loss of the best rank-$k$ approximation of $A$ returned by $S_g$ by at least a constant factor.*

*Proof:*

$$\mathbf{E}_{S_r}\left[\min_{\text{rank-}k\ X \in \text{rowsp}(S_r A)} \|X - A\|_F^2\right] \geq n + \frac{\ell k}{4e} - (1 + \alpha)k - n^{-O(1)} \qquad\qquad \triangleright \text{Lemma C.12}$$

$$\geq (1 + 1/\alpha)(n - s) \qquad\qquad\qquad \triangleright \ell \geq \frac{(4e+1)n}{\alpha k}$$

$$= (1 + 1/\alpha) \min_{\text{rank-}k\ X \in \text{rowsp}(S_g A)} \|X - A\|_F^2 \qquad \triangleright \text{Corollary C.11}$$

$$\square \quad \blacksquare$$

## C.2 ZIPFIAN ON SQUARED ROW NORMS.

Each matrix $A \in \mathbb{R}^{n \times d} \sim \mathcal{A}_{zipf}$ has rows which are uniformly random and orthogonal. Each $A$ has $2^{i+1}$ rows of squared norm $n^2/2^{2i}$ for $i \in [1, \ldots, O(\log(n))]$. We also assume that each row has the same squared norm for all members of $\mathcal{A}_{zipf}$.

In this section, the $s$ rows with largest norm are called the *heavy* rows and the remaining are the *light* rows. For convenience, we number the heavy rows $1 - s$; however, the heavy rows can appear at any indices, as long as any row of a given index has the same norm for all members of $\mathcal{A}_{zipf}$. Also, we assume that $s \leq k/2$ and, for simplicity, $s = \sum_{i=1}^{h_s} 2^{i+1}$ for some $h_s \in \mathbb{Z}^+$. That means the minimum squared norm of a heavy row is $n^2/2^{2h_s}$ and the maximum squared norm of a light row is $n^2/2^{2h_s+2}$.

The analysis of the greedy algorithm ordered by non-increasing row norms on this family of matrices is similar to our analysis for the spiked covariance model. Here we analyze the case in which rows are orthogonal. By continuity, if the rows are close enough to being orthogonal, all decisions made by the greedy algorithm will be the same.

As a first step, by Lemma C.6, at the end of iteration $k$ the first $k$ rows are assigned to different bins. Then, via a similar inductive proof, we show that none of the light rows are mapped to a bin that contains one of the top $s$ heavy rows.

**Lemma C.15.** *At each iteration $k + r$, the greedy algorithm picks the position of the non-zero value in the $(k + r)$-th column of the CountSketch matrix $S$ so that the light row $A_{k+r}$ is mapped to a bin that does not contain any of top $s$ heavy rows.*

*Proof:* We prove the statement by induction. The base case $r = 0$ trivially holds as the first $k$ rows are assigned to distinct bins. Next we assume that in none of the first $k + r - 1$ iterations a light row is assigned to a bin that contains a heavy row. Now, we consider the following cases:

**1. If $A_{k+r}$ is assigned to a bin that only contains light rows.** Without loss of generality we can assume that $A_{k+r}$ is assigned to $b_k$. Since the vectors are orthogonal, we only need to bound the difference in the projection of $A_{k+r}$ and the light rows that are assigned to $b_k$ onto the direction of $w_k$ before and after adding $A_{k+r}$ to $b_k$. In this case, the total squared loss corresponding to rows in $b_k$ and $A_{k+r}$ before and after adding $A_{k+1}$ are respectively

$$\text{before adding } A_{k+r} \text{ to } b_k: \|A_{k+r}\|_2^2 + \sum_{A_j \in b_k} \|A_j\|_2^2 - \left(\frac{\sum_{A_j \in b_k} \|A_j\|_2^4}{\sum_{A_j \in b_k} \|A_j\|_2^2}\right)$$

$$\text{after adding } A_{k+r} \text{ to } b_k: \|A_{k+r}\|_2^2 + \sum_{A_j \in b_k} \|A_j\|_2^2 - \left(\frac{\|A_{k+r}\|_2^4 + \sum_{A_j \in b_k} \|A_j\|_2^4}{\|A_{k+r}\|_2^2 + \sum_{A_j \in b_k} \|A_j\|_2^2}\right)$$

Thus, the amount of increase in the squared loss is

$$
\left(\frac{\sum_{A_j \in b_k} \|A_j\|_2^4}{\sum_{A_j \in b_k} \|A_j\|_2^2}\right) - \left(\frac{\|A_{k+r}\|_2^4 + \sum_{A_j \in b_k} \|A_j\|_2^4}{\|A_{k+r}\|_2^2 + \sum_{A_j \in b_k} \|A_j\|_2^2}\right) = \frac{\|A_{k+r}\|_2^2 \cdot \sum_{A_j \in b_k} \|A_j\|_2^4 - \|A_{k+r}\|_2^4 \cdot \sum_{A_j \in b_k} \|A_j\|_2^2}{(\sum_{A_j \in b_k} \|A_j\|_2^2)(\|A_{k+r}\|_2^2 + \sum_{A_j \in b_k} \|A_j\|_2^2)}
$$

$$
= \|A_{k+r}\|_2^2 \cdot \frac{\frac{\sum_{A_j \in b_k} \|A_j\|_2^4}{\sum_{A_j \in b_k} \|A_j\|_2^2} - \|A_{k+r}\|_2^2}{\sum_{A_j \in b_k} \|A_j\|_2^2 + \|A_{k+r}\|_2^2}
$$

$$
\leq \|A_{k+r}\|_2^2 \cdot \frac{\sum_{A_j \in b_k} \|A_j\|_2^2 - \|A_{k+r}\|_2^2}{\sum_{A_j \in b_k} \|A_j\|_2^2 + \|A_{k+r}\|_2^2}
$$

$$\tag{C.7}$$

**2. If $A_{k+r}$ is assigned to a bin that contains a heavy row.** Without loss of generality and by the induction hypothesis, we assume that $A_{k+r}$ is assigned to a bin $b$ that only contains a heavy row $A_j$. Since the rows are orthogonal, we only need to bound the difference in the projection of $A_{k+r}$ and $A_j$ In this case, the total squared loss corresponding to $A_j$ and $A_{k+r}$ before and after adding $A_{k+1}$

to $b$ are respectively

$$\text{before adding } A_{k+r} \text{ to } b_k\text{: } \|A_{k+r}\|_2^2$$

$$\text{after adding } A_{k+r} \text{ to } b_k\text{: } \|A_{k+r}\|_2^2 + \|A_j\|_2^2 - (\frac{\|A_{k+r}\|_2^4 + \|A_j\|_2^4}{\|A_{k+r}\|_2^2 + \|A_j\|_2^2})$$

Thus, the amount of increase in the squared loss is

$$\|A_j\|_2^2 - (\frac{\|A_{k+r}\|_2^4 + \|A_j\|_2^4}{\|A_{k+r}\|_2^2 + \|A_j\|_2^2}) = \|A_{k+r}\|_2^2 \cdot \frac{\|A_j\|_2^2 - \|A_{k+r}\|_2^2}{\|A_j\|_2^2 + \|A_{k+r}\|_2^2} \tag{C.8}$$

Then (C.8) is larger than (C.7) if $\|A_j\|_2^2 \geq \sum_{A_i \in b_k} \|A_i\|_2^2$. Next, we show that at every inductive iteration, there exists a bin $b$ which only contains light rows and whose squared norm is smaller than the squared norm of any heavy row. For each value $m$, define $h_m$ so that $m = \sum_{i=1}^{h_m} 2^{i+1} = 2^{h_m+2} - 2$.

Recall that all heavy rows have squared norm at least $\frac{n^2}{2^{2h_s}}$. There must be a bin $b$ that only contains light rows and has squared norm at most

$$\|w\|_2^2 = \sum_{A_i \in b} \|A_i\|_2^2 \leq \frac{n^2}{2^{2(h_s+1)}} + \frac{\sum_{i=h_k+1}^{h_n} \frac{2^{i+1}n^2}{2^{2i}}}{k-s}$$

$$\leq \frac{n^2}{2^{2(h_s+1)}} + \frac{2n^2}{2^{h_k}(k-s)}$$

$$\leq \frac{n^2}{2^{2(h_s+1)}} + \frac{n^2}{2^{2h_k}} \qquad \triangleright s \leq k/2 \text{ and } k > 2^{h_k+1}$$

$$\leq \frac{n^2}{2^{2h_s+1}} \qquad \triangleright h_k \geq h_s + 1$$

$$< \|A_s\|_2^2$$

Hence, the greedy algorithm will map $A_{k+r}$ to a bin that only contains light rows. $\qquad \square$

**Corollary C.16.** *The squared loss of the best* $\mathrm{rank}$-$k$ *approximate solution in the rowspace of* $S_g A$ *for* $A \in \mathbb{R}^{n \times d} \sim \mathcal{A}_{zipf}$ *where* $A \in \mathbb{R}^{n \times d}$ *and* $S_g$ *is the CountSketch constructed by the greedy algorithm with non-increasing order, is* $< \frac{n^2}{2^{h_k-2}}$.

*Proof:* At the end of iteration $k$, the total squared loss is $\sum_{i=h_k+1}^{h_n} 2^{i+1} \cdot \frac{n^2}{2^{2i}}$. After that, in each iteration $k + r$, by (C.7), the squared loss increases by at most $\|A_{k+r}\|_2^2$. Hence, the total squared loss in the solution returned by $S_g$ is at most

$$2(\sum_{i=h_k+1}^{h_n} \frac{2^{i+1}n^2}{2^{2i}}) = 4n^2 \cdot \sum_{i=h_k+1}^{h_n} \frac{1}{2^i}$$

$$< \frac{4n^2}{2^{h_k}} = \frac{n^2}{2^{h_k-2}} \qquad \square$$

Next, we bound the squared loss of the best $\mathrm{rank}$-$k$-approximate solution constructed by the standard CountSketch with a randomly chosen sparsity pattern.

**Observation C.17.** *Let us assume that the orthogonal rows* $A_{r_1}, \cdots, A_{r_c}$ *are mapped to same bin and for each* $i \leq c$, $\|A_{r_1}\|_2^2 \geq \|A_{r_i}\|_2^2$. *Then, the total squared loss of* $A_{r_1}, \cdots, A_{r_c}$ *after projecting onto* $A_{r_1} \pm \cdots \pm A_{r_c}$ *is at least* $\|A_{r_2}\|_2^2 + \cdots + \|A_{r_c}\|_2^2$.

*Proof:* Note that since $A_{r_1}, \cdots, A_{r_c}$ are orthogonal, for each $i \leq c$, the squared projection of $A_{r_i}$ onto $A_{r_1} \pm \cdots \pm A_{r_c}$ is $\|A_{r_i}\|_2^4 / \sum_{j=1}^c \|A_{r_j}\|_2^2$. Hence, the sum of squared projection coefficients of $A_{r_1}, \cdots, A_{r_c}$ onto $A_{r_1} \pm \cdots \pm A_{r_c}$ is

$$\frac{\sum_{j=1}^c \|A_{r_j}\|_2^4}{\sum_{j=1}^c \|A_{r_j}\|_2^2} \leq \|A_{r_1}\|_2^2$$

Hence, the total projection loss of $A_{r_1}, \cdots, A_{r_c}$ onto $A_{r_1} \pm \cdots \pm A_{r_c}$ is at least

$$\sum_{j=1}^{c} \|A_{r_j}\|_2^2 - \|A_{r_1}\|_2^2 = \|A_{r_2}\|_2^2 + \cdots + \|A_{r_c}\|_2^2. \qquad \square$$

In particular, Observation C.17 implies that whenever two rows are mapped into a same bin, the squared norm of the row with smaller norm *fully* contributes to the total squared loss of the solution.

**Lemma C.18.** *For $k > 2^{10} - 2$, the expected squared loss of the best* rank-$k$ *approximate solution in the rowspace of $S_r A$ for $A_{n \times d} \sim \mathcal{A}_{zipf}$, where $S_r$ is the sparsity pattern of a CountSketch chosen uniformly at random, is at least $\frac{1.095 n^2}{2^{h_k - 2}}$.*

*Proof:* In light of Observation C.17, we need to compute the expected number of collision between rows with "large" norm. We can interpret the randomized construction of the CountSketch as a "balls and bins" experiment.

For each $0 \le j \le h_k$, let $\mathcal{A}_j$ denote the set of rows with squared norm $\frac{n^2}{2^{2(h_k - j)}}$ and let $\mathcal{A}_{>j} = \bigcup_{j < i \le h_k} \mathcal{A}_i$. Note that for each $j$, $|\mathcal{A}_j| = 2^{h_k - j + 1}$ and $|\mathcal{A}_{>j}| = \sum_{i=j+1}^{h_k} 2^{h_k - i + 1} = \sum_{i=1}^{h_k - j} 2^i = 2(2^{h_k - j} - 1)$. Moreover, note that $k = 2(2^{h_k + 1} - 1)$. Next, for a row $A_r$ in $\mathcal{A}_j$ ($0 \le j < h_k$), we compute the probability that at least one row in $\mathcal{A}_{>j}$ collide with $A_r$.

$$\mathbf{Pr}[\text{at least one row in } \mathcal{A}_{>j} \text{ collide with } A_r] = (1 - (1 - \frac{1}{k})^{|\mathcal{A}_{>j}|})$$

$$\ge (1 - e^{-\frac{|\mathcal{A}_{>j}|}{k}})$$

$$= (1 - e^{-\frac{2^{h_k - j} - 1}{2^{h_k + 1} - 1}})$$

$$\ge (1 - e^{-2^{-j-2}}) \qquad \triangleright \text{ since } \frac{2^{h_k - j} - 1}{2^{h_k + 1} - 1} > 2^{-j-2}$$

Hence, by Observation C.17, the contribution of rows in $\mathcal{A}_j$ to the total squared loss is at least

$$(1 - e^{-2^{-j-2}}) \cdot |\mathcal{A}_j| \cdot \frac{n^2}{2^{2(h_k - j)}} = (1 - e^{-2^{-j-2}}) \cdot \frac{n^2}{2^{h_k - j - 1}}$$

$$= (1 - e^{-2^{-j-2}}) \cdot \frac{n^2}{2^{h_k - 2}} \cdot 2^{j-1}$$

Thus, the contribution of rows with "large" squared norm, i.e., $\mathcal{A}_{>0}$, to the total squared loss is at least[5]

$$\frac{n^2}{2^{h_k - 2}} \cdot \sum_{j=0}^{h_k} 2^{j-1} \cdot (1 - e^{-2^{-j-2}}) \ge 1.095 \cdot \frac{n^2}{2^{h_k - 2}} \qquad \triangleright \text{for } h_k > 8 \qquad \square$$

**Corollary C.19.** *Let $S_g$ be a CountSketch whose sparsity pattern is learned over a training set drawn from $\mathcal{A}_{sp}$ via the greedy approach. Let $S_r$ be a CountSketch whose sparsity pattern is picked uniformly at random. Then, for an $n \times d$ matrix $A \sim \mathcal{A}_{zipf}$, for a sufficiently large value of $k$, the expected loss of the best* rank-$k$ *approximation of $A$ returned by $S_r$ is worse than the approximation loss of the best* rank-$k$ *approximation of $A$ returned by $S_g$ by at least a constant factor.*

*Proof:* The proof follows from Lemma C.18 and Corollary C.16. $\qquad \square$

**Remark C.20.** We have provided evidence that the greedy algorithm that examines the rows of $A$ according to a non-increasing order of their norms (i.e., *greedy with non-increasing order*) results in a better rank-$k$ solution compared to the CountSketch whose sparsity pattern is chosen at random. However, still other implementations of the greedy algorithm may result in a better solution compared to the greedy with non-increasing order. To give an example, in the following simple instance the greedy algorithm that check the rows of $A$ in a random order (i.e., *greedy with random order*) achieves a rank-$k$ solution whose cost is a constant factor better than the solution returned by the greedy with non-increasing order.

---

[5]The numerical calculation is computed by WolframAlpha.

Let $A$ be a matrix with four orthogonal rows $u, u, v, w$ where $\|u\|_2 = 1$ and $\|v\|_2 = \|w\|_2 = 1 + \epsilon$ and suppose that the goal is to compute a rank-2 approximation of $A$. Note that in the greedy with non-decreasing order, $v$ and $w$ will be assigned to different bins and by a simple calculation we can show that the copies of $u$ also will be assigned to different bins. Hence, the squared loss in the computed rank-2 solution is $1 + \frac{(1+\epsilon)^2}{2+(1+\epsilon)^2}$. However, the optimal solution will assign $v$ and $w$ to one bin and the two copies of $u$ to the other bin which results in the squared loss of $(1 + \epsilon)^2$ which is a constant factor smaller than the solution returned by the greedy with non-increasing order for sufficiently small values of $\epsilon$.

On the other hand, in the greedy algorithm with random order, with a constant probability $(\frac{1}{3} + \frac{1}{8})$, the computed solution is the same as the optimal solution. Otherwise, the greedy algorithm a with random order returns the same solution as the greedy algorithm with a non-increasing order. Hence, in expectation, the solution returned by the greedy with random order is better than the solution returned by the greedy algorithm with non-increasing order by a constant factor.

# D    EXPERIMENTS - APPENDIX

## D.1    BASELINES

We comment on two of our baselines:

- **Exact SVD**: In the canonical, learning-free sketching setting (i.e., any matrix is equally probable), sketching using the top $m$ singular vectors yields a $(1 + \epsilon)$-approximation for both LRA and $k$-means Cohen et al. (2015).
- **Column sampling**: In the canonical, learning-free sketching setting, sketching via column sampling yields a $(1 + \epsilon)$-approximation for $k$-means Cohen et al. (2015).

## D.2    EXPERIMENTAL PARAMETERS

For the tables in Section 5, we describe experimental parameters. First, we provide some general implementation details.

We implemented both the greedy (Algorithm 1) and stochastic gradient descent (Algorithm 2) algorithms in PyTorch. In the first case, PyTorch allowed us to harness GPUs to speed up computation on large matrices. We used several Nvidia GeForce GTX 1080 Ti machines. In the second case, PyTorch allowed us to effortlessly compute numerical gradients for each task's objective function. Specifically, PyTorch provides automatic differentiation, which is implemented by backpropagation through chained differentiable operators.

There are also two points of note in the greedy algorithm implementation. First, we noticed that for MRR and LRA, each iteration required computing the SVD for many rank-1 updates of the current $S$. Instead of computing the SVD from scratch for each of these variants, we first computed the SVD of $S$ and then used fast rank-1 SVD updates Brand (2006). This greatly improved the runtime of Algorithm 1. Second, we decided to set $\mathcal{D}_w$ (the set of candidate row weights) to 10 samples in $[-2, 2]$ because we noticed most weights were in this range after running Algorithm 2.

## D.3    RUNNING TIME

We examine the runtimes of our various sketching algorithms. In Table D.1, the times are obtained for the LRA task with $k = 30, m = 60$ on the logo data set. However, similar trends should be expected for other combinations of task, task parameters, and data sets.

We define the *inference* runtime as the time to apply the sketching algorithm. The *training* runtime is the time to train a sketch on $\mathcal{A}_{\text{train}}$ and only applies to learned sketches. Generally, the long *training* times are not problematic because training is only done once and can be completed offline. On the other hand, the *inference* runtime should be as fast as possible.

Note that *inference* was timed using 1 matrix from $\mathcal{A}_{\text{test}}$ on an Nvidia Geforce GTX 1080 Ti GPU. The values were averaged over 10 trials.

We observe that sparse sketches (such as the ones used in *GD only* and *Ours*) have much lower *inference* runtimes than the dense sketches of *exact SVD*.

Table D.1: Timing comparison of sketching algorithms for LRA (using Algorithm 1 from Indyk et al. (2019))

|  | Time (sec) |
|---|---|
| **random: inference** | 0.0114 |
| **exact SVD: inference** | 0.185 |
| **learned (random pattern): training** | 193 (3 min) |
| **learned (random pattern): inference** | 0.0114 |
| **learned (greedy pattern): training** | 6300 (1 h 45 min) |
| **learned (greedy pattern): inference** | 0.0114 |

