# OpenReview forum: "A framework for learned CountSketch"
_ICLR.cc/2021/Conference — Reject_

### Official Review · AnonReviewer2 · 2020-10-28
**Recommendation to accept**

**Rating:** 7
**Confidence:** 4

**Review:**

Summary:

The authors consider the problem of ``"sketching" – a popular compression technique in machine learning - used for reducing the size of the data enabling one to quickly compute an approximate solution using this compressed input. This paper introduces a general framework for learning and applying sparse sketching matrices. A two-stage procedure for learning sketches with the same sparsity pattern as CountSketch is proposed which involves first learning the sketch’s non-zero entries, and then optimizing their values. They then show how to apply the obtained sketch so that it has worst case approximation error guarantees. This procedure is applied to three applications, namely least squares regression, low rank approximation (LRA) and k-means clustering. Experimental results demonstrate a substantial reduction in the approximation error compared to other baseline approaches (including classically learned sketches). On the theoretical front, it is shown for regression and LRA that the proposed approach obtains improved error guarantees for fixed time complexity.  Additionally, it is shown for LRA (under certain input distributions) that including the first stage is strictly better than not including it. Finally, a more straightforward way of retaining worst case approximation guarantees for k-means is shown.

----------------------------------------------------------
Reasons for score:

Overall, I vote for accepting the paper. The procedure for learning and applying the sketching matrix seems novel and interesting, and the empirical results (Especially for LRA and least squares regression) on real datasets suggest that the proposed method does substantially improve over other baseline methods, including the classical CountSketch. The theoretical results for least squares regression and LRA also imply that their algorithm achieves a better accuracy than the classical CountSketch approach (given the same time complexity). I do have some questions mentioned in “Cons” and “Further remarks” below which I would like the authors to clarify in the rebuttal phase.

------------------------------------------------------------------------------------
Pros:

-	Paper is in general written well barring some notational issues and explanations which were not clear to me (explained later in “Cons”)
-	The two-stage idea of learning a sketching matrix with the same sparsity pattern as in CountSketch is interesting. Instantiating the framework to problems such as: least squares regression, LRA and k-means is done nicely, and the theoretical results appear to be strong.
-	The empirical evaluation on real datasets makes a convincing case for the proposed algorithm and demonstrates that learning the CountSketch matrix can indeed be substantially better than using the classical CountSketch, on real data.

----------------------------------------------------------------------------------
Cons:

-	The sketching matrices considered in the paper are restricted to Count Sketches which have a particular sparsity pattern. So, the framework is not as general as is claimed in the abstract.
-	Some of the notation and explanations in Section 3 are quite unclear. For example, the loss function symbol is sometimes used as L(X), sometimes as L(S, I) etc. In Algorithm 1, I don’t understand what range(n) means, and also the summation over A_i \in A_{train} seems strange. Algorithms 1,2 have a training set \mathcal{A}_{train} as input but I don’t see how this appears for the examples in Section 4; is N =1 there? In stage 2, what does the operation S[p, range(n)] = v do? In Algorithm 3, it would be helpful to explain (with an example perhaps) what the symbol \mathcal{M}(S,I) could look like. At the beginning of page 4, it is mentioned that learned sketches can encounter out of distribution inputs etc. But some of the guarantees in Section 4 (Theorems 4.1,4.2) apply for arbitrary inputs, isn’t it?
-	Theorems 4.1, 4.2 assume that the learned sketch matrices are affine \beta embeddings and hence attain a (1+\beta) approximation for the respective problems. But it is not mentioned whether this assumption can be satisfied.

---------------------------------------------------------------------
Further remarks:

-	I don’t understand why in page 2 (“Our results” para) it is written that the sketches are input-independent, aren’t the sketches derived in the paper a function of the input data (for e.g. in Alg. 1)?
-	For theorems 4.1 and 4.2, it will be helpful to state the corresponding guarantee for the classical algorithm in the main text to enable a direct comparison.
-	In the abstract and the introduction, it is mentioned that for LRA it strictly better to include the first optimization stage. But this is shown for a couple of specific input distributions, hence should be clearly specified as such.
-	In Algorithms 1,2, sometimes "\mathcal{A}_{train}" is used and sometimes "A_{train}". If sets are being denoted with calligraphic symbols, then mention this in the notations paragraph.

----------------------------------------------------------
Post rebuttal: I have read the authors response and will go with my original score for this paper.

---

> ### Author Response · Authors · 2020-11-25
> **Thank you for your feedback**
>
> Thank you very much for your careful and detailed response.
>
> 1. It is true that this framework is for CountSketch matrices rather than general sparse matrices. We updated the paper and title accordingly.
> 2. We've updated the notation, which is clearer and consistent throughout the paper now. In Algorithm 1, $range(n)$ is $1,\ldots,n$; it is used to iterate over the columns of CountSketch S. In the updated paper, we explain at the top of p. 4 that we average/sum over $A_i \in A_{train}$ because we are performing empirical risk minimization (ERM). $A_{train}$ does not appear in Section 4, because Section 4 defines functions of an individual matrix $A$; in Section 2, we sum over such functions for $A \in A_{train}$.
> In Stage 2, we replaced $S[p, range(n)] = v$ with a clearer explanation; essentially, it constructs a CountSketch $S$ from a condensed $(p, v)$ representation. In Alg. 3, we have defined and illustrated $M$. In the section "Learned sketch with worst-case guarantees", we explained why our method provides worst-case guarantees against arbitrary inputs, including out-of-distribution inputs.
> 3. Regarding the beta embedding: we have updated the paper with a more precise description of our results (Theorems 4.1 and 4.3). In particular, we have added Remarks 4.2 and 4.4 to clarify the relationship between Algorithm 3's performance, the approximation errors of the approximate solutions computed using learned and classical sketches, and the trade-off parameter $\beta$.
> 4. The definition of "input-independent" sketches: while learned CountSketch is data-dependent (it is optimized using sample input matrices), it is still considered input-independent in our notation because it is applied to unseen input matrices (test samples).
> 5. We have noted the input distribution assumption for our LRA proof.
> 6. We have also fixed all inconsistencies in our notation. Thanks for pointing them out!

---

### Official Review · AnonReviewer1 · 2020-10-30
**The idea of learning sketches and the proposed greedy algorithms are simple. More numerical results on different datasets are needed to exemplify.**

**Rating:** 6
**Confidence:** 2

**Review:**

This paper introduces a general framework for learning and applying input-independent sparse sketches.  Then this paper instantiates this framework with three sketching applications: least-squares regression, low-rank approximation (LRA), and k-means clustering. Numerical experiments are given to demonstrate that the approach substantially decreases approximation error compared to classical and naively learned sketches.

The idea of learning sketches and the proposed greedy algorithms are simple. They are closely related to some of the existing related works on dictionary learning. The numerical results look promising. I would suggest to give more numerical results on different datasets to exemplfy the proposed approach, as that the main contribution of the paper is on algorithmic aspect.

---

> ### Author Response · Authors · 2020-11-25
> **Thank you for your feedback**
>
> Thank you very much for your careful and detailed response.
>
> We have added numerical results on two more data sets for a total of five.
> We now have a range of data set *types*: image, text, and graphs (adjacency matrices).
>
> In (sparse) dictionary learning, the goal is to represent a given set of observations $M\in \mathbb{R}^{d \times k}$ in the form of $D R$, where the representation $R\in \mathbb{R}^{n\times k}$ is a sparse matrix.
> However, here we use sparse matrices to compute solutions to problems of interest, such as, LRA, MRR, and k-means clustering, more efficiently. In particular, the solutions we compute via sparse sketching matrices are not necessarily sparse themselves. Another way of thinking of our problems in terms of dictionary learning could be to capture $AS$ in our applications, where $S$ is a sparse sketching matrix and $A$, as the input matrix $DR$ in dictionary learning. With this point of view, in our application, we need to find $M$ given $DR$, while in dictionary learning $M$ is known and the goal is to find $DR$. Considering both of these interpretations, we believe that our main focus in this paper is different from the goal in dictionary learning.

---

### Official Review · AnonReviewer5 · 2020-11-07
**Novelty unclear**

**Rating:** 5
**Confidence:** 3

**Review:**

The authors consider a specific sketching method, CountSketch, and three objective functions defined over the data design matrix: multiple-response regression (MMR), low-rank approximation (LRA), and k-means clustering. They compare the classical CountSketch with a random choices of the {-1,+1}-valued sketching matrix against: (1) Gradient descent optimization of the CountSketch weights. This was previously introduced for LRA and the authors extend it to MMR and k-means. (2) Greedy optimization of the positions of the CountSketch non-zero entries.

In order to avoid losing the worst-case guarantees of the original (non-optimized) sketch, a wrapper algorithm guarantees that the optimized sketch is considered only if it is not worse than the optimized sketch.

There are theoretical and empirical results. The first two theoretical results show that for MMR and LRA the wrapper algorithm can be implemented without having to solve the problem twice on the full dataset. Theorem 4.3 shows that greedy optimization produces a strictly better result for the LRA objective than CountSketch alone when data are sampled from a certain distribution. Theorem 4.4 finally shows that a different wrapping technique (used in a previous result for LRA), can be applied to the k-means objective.

Experiments are performed on three real-world datasets comparing CountSketch, gradient optimization, and greedy+gradient optimization on the three objectives. The bottom line is that the greedy optimization step helps significantly.

The paper is not an easy read and the clarity can be significantly improved. There is not a lot of conceptual novelty compared to previous works: just the introduction of the greedy optimization step. There is technical work to extend gradient optimization and prove the theoretical results. However, the extent of the overall technical novelty remains unclear.

It is not explicitely written that the greedy optimization step is performed only on the training part of the data.

Is greedy+gradient optimization fully deterministic?

Standard deviations are not reported in the expertiments.

---

> ### Author Response · Authors · 2020-11-25
> **Thank you for the feedback**
>
> Thank you very much for your careful and detailed response.
>
> There are several conceptual contributions in our paper, namely:
> 1. Using "*fast* comparisons" to retain worst-case guarantees. Although simple, a previous work took another approach ("sketch concatenation") which is application-specific and only proven for LRA. Our method is far more general.
> 2. That said, "sketch concatenation" is slightly simpler. So, we expanded its applicability and proved that it works for another task, k-means.
> 3. The notion of *sparsity pattern* optimization, which is crucial for *sparse* matrices. This alone gives a 12% and 20% increase in accuracy for LRA and MRR, respectively.
>
> At a high level, we have made the sketch learning process more general and effective.
>
> We also made several important theoretical contributions:
> 1. We formulated a time-optimal algorithm for *learned sketching with guarantees*. This was challenging because our method for sketching with guarantees had to attain the same time complexity as vanilla sketching.
> 2. *k-means with guarantees*: We proved that the "sketch concatenation" method (sketching with the concatenation of a classical and a learned sketch) can be used for k-means to obtain worst-case accuracy guarantees.
> More specifically, we proved that classical sketches which satisfy a "strong coreset" property (such as CountSketch) also satisfy a "sketch monotonicity" property for k-means, which says that any extension to a sketch provides a better k-means solution. This result may be of independent interest.
> 3. *Greedy step analysis*: We proved that under certain assumptions (on the input distribution, matrix size, etc.), a greedy-initialized sketch is strictly better than a randomly initialized sketch. This is a new result which leverages the following insight:
>
>     the top k-rank subspace of a matrix's row span is mostly "covered" by the k rows with largest norm. Since sketches with a single non-zero entry per column can be viewed as hashing rows together, it is crucial for the hash to separate these largest norm rows so that their directions are preserved. The greedy algorithm essentially ensures this through its placement of the sketch's nonzero entries.
>
>
> Finally, we would like to add that the simplicity of our approach is an advantage, not a shortcoming. Simpler systems are more likely to be adopted. Our system has clear real-world applications, which we highlighted in the experiments. We had excellent empirical results on
> 1. LRA and k-means on images, which has compression applications
> 2. SVD/LRA on text data, which has NLP applications (namely, Latent Semantic Indexing)
> 3. LRA on graph adjacency matrices, which has applications to approximately computing max cuts
>
> Further, there is now more mainstream interest in sketching. NVIDIA has [recently looked into](https://developer.download.nvidia.com/video/gputechconf/gtc/2019/presentation/s9226-fast-singular-value-decomposition-on-gpus-v2.pdf) implementing approximate SVD (via sketching) for CUDA, their parallel computing platform. For these reasons, this approach is likely to be practically applicable.
>
> **Answers to questions**
> 1. Greedy optimization is only performed on the training data, as mentioned in Algorithm
> 2. Greedy and gradient optimization is not deterministic. There are two sources of randomness: mini-batch sampling and row order randomization for greedy. However, our results can be replicated by using our random seeds and hardware.
> 3. Standard deviations are now reported, along with commentary.
> 4. Also, we revised the Introduction and Framework sections to improve clarity. They now both have better organization and articulation. Also, the Framework section has improved notation and an illustrative example.

---

### Comment · ~Simin_Liu1 · 2021-02-24
**New version available**

Recently, we made some significant updates and improvements. The new version can be found [here](https://arxiv.org/abs/2007.09890).

The main differences are:
1. Expanded theoretical results: we provide two modifications to classical sketching algorithms that allow the use of learned sketches with worst-case guarantees. We prove one works for all problems, and now also prove that the other works for LRA and k-means. These results are the *first time-optimal learned sketch algorithms*.
2. A more thorough look at the algorithm: we explain why it's an important improvement and how we used problem-specific strategies to make it more tractable. We also prove that it is better than the random algorithm for LRA under certain conditions.
3. Restructured content, with added commentary

Thanks to the reviewers for their helpful remarks.

---

### Decision · Program_Chairs · 2021-01-07
**Final Decision**

**Decision:**

Reject

**Comment:**

The work falls under the setting of learning-based sketching/compressive subsampling. It extends the work of Indyk et al 2019 (including sparsity pattern optimization and some theoretical enhancements).  The reviewers agree that while the conceptual novelty including the greedy optimization step is not too much, it is nonetheless interesting and is non-trivial. However, given the highly competitive submissions at ICLR, the current scores are not sufficient for acceptance.